

COMPUTO

ISSN 2824-7795

# Efficient simulation of individual-based population models

## The `R` package `IBMPopSim`

Daphné Giorgi[1]    Laboratoire de Probabilités, Statistique et Modélisation, Sorbonne Université, CNRS

Sarah Kaakai[2]    Laboratoire Manceau de Mathématiques, Le Mans Université

Centre de Mathématiques Appliquées, Ecole Polytechnique, CNRS

Vincent Lemaire [3]    Laboratoire de Probabilités, Statistique et Modélisation, Sorbonne Université, CNRS

Date published: 2024-10-11    Last modified: 2024-10-11

## Abstract

The `R` Package `IBMPopSim` facilitates the simulation of the random evolution of heterogeneous populations using stochastic Individual-Based Models (IBMs). The package enables users to simulate population evolution, in which individuals are characterized by their age and some characteristics, and the population is modified by different types of events, including births/arrivals, death/exit events, or changes of characteristics. The frequency at which an event can occur to an individual can depend on their age and characteristics, but also on the characteristics of other individuals (interactions). Such models have a wide range of applications in fields including actuarial science, biology, ecology or epidemiology. `IBMPopSim` overcomes the limitations of time-consuming IBMs simulations by implementing new efficient algorithms based on thinning methods, which are compiled using the `Rcpp` package while providing a user-friendly interface.

*Keywords:* Individual-based models, stochastic simulation, population dynamics, Poisson measures, thinning method, actuarial science, insurance portfolio simulation

# Contents

[1]Corresponding author: daphne.giorgi@sorbonne-universite.fr

[2]Corresponding author: sarah.kaakai@univ-lemans.fr

[3]Corresponding author: vincent.lemaire@sorbonne-universite.fr

# 1 Introduction

In various fields, advances in probability have contributed to the development of a new mathematical framework for so-called individual-based stochastic population dynamics, also called stochastic Individual-Based Models (IBMs). Stochastic IBMs allow the modeling in continuous time of populations dynamics structured by age and/or characteristics. In the field of mathematical biology and ecology, a large community has used this formalism for the study of the evolution of structured populations (see e.g. Ferrière and Tran (2009); Collet, Méléard, and Metz (2013); Bansaye and Méléard

(2015); Costa et al. (2016); Billiard et al. (2016); Lavallée et al. (2019); Méléard, Rera, and Roget (2019); Calvez et al. (2020)), after the pioneer works (Fournier and Méléard 2004; Tran 2008; Méléard and Tran 2009). IBMs are also useful in demography and actuarial sciences, for the modeling of human populations dynamics (see e.g. Bensusan (2010); Boumezoued (2016); El Karoui, Hadji, and Kaakai (2021)).

Indeed, they allow the modeling of heterogeneous and complex population dynamics, which can be used to compute demographic indicators or simulate the evolution of insurance portfolios in order to study the basis risk, compute cash flows for annuity products or pension schemes, or for a fine assessment of mortality models (Barrieu et al. 2012). There are other domains in which stochastic IBMs can be used, for example in epidemiology with stochastic compartmental models, neurosciences, cyber risk, or Agent-Based Models (ABMs) in economy and social sciences, which can be seen as IBMs. Many mathematical results have been obtained in the literature cited above, for quantifying the limit behaviors of IBMs over long time scales or in large population. In particular, pathwise representations of IBMs have been introduced in Fournier and Méléard (2004) (and extended to age-structured populations in Tran (2008); Méléard and Tran (2009)), as measure-valued pure jumps Markov processes, solutions of SDEs driven by Poisson measures. These pathwise representations are based on the *thinning* and projection of Poisson random measures defined on extended spaces. However, the simulation of large and interacting populations is often computationally expensive.

The aim of the R package IBMPopSim is to meet the needs of the various communities for efficient tools in order to simulate the evolution of stochastic IBMs. IBMPopSim provides a general framework for the simulation of a wide class of IBMs, where individuals are characterized by their age and/or a set of characteristics. Different types of events can be included in the modeling by users, depending on their needs: births, deaths, entry or exit in/to the population and changes of characteristics (swap events). Furthermore, the various events that can happen to individuals in the population can occur at a non-stationary frequency, depending on the individuals' characteristics and time, and also including potential interactions between individuals.

We introduce a unified mathematical and simulation framework for this class of IBMs, generalizing the pathwise representation of IBMs by thinning of Poisson measures, as well as the associated population simulation algorithm, based on an acceptance/rejection procedure. In particular, we provide general sufficient conditions on the event intensities under which the simulation of a particular model is possible.

We opted to implement the algorithms of the IBMPopSim package using the Rcpp package, a tool facilitating the seamless integration of high-performance C++ code into easily callable R functions (Eddelbuettel and Francois 2011). IBMPopSim offers user-friendly R functions for defining and simulating IBMs. Once events and their associated intensities are specified, an automated procedure creates the model. This involves integrating the user's source code into the primary C++ code using a template mechanism. Subsequently, Rcpp is invoked to compile the model so that the model is integrated into the R session and callable with varying parameters, enabling the generation of diverse population evolution scenarios. Combined with the design of the simulation algorithms, the package structure yields very competitive simulation runtimes for IBMs, while staying user-friendly for R users. Several outputs function are also implemented in IBMPopSim. For instance the package allows the construction and visualization of age pyramids, as well as the construction of death and exposures table from the censored individual data, compatible with R packages concerned with mortality modelling, such as Hyndman et al. (2023) or A. Villegas, Millossovich, and Kaishev Hyndman (2018). Several examples are provided in the form of R vignettes on the website, and in recent works of El Karoui, Hadji, and Kaakai (2021) and Roget et al. (2024).

To the best of our knowledge, there are no other R packages currently available addressing the issue

of stochastic IBMs efficient simulation. Another approach for simulating populations is continuous time microsimulation in social sciences, which is implemented in the R package MicSim (Zinn 2014). In this framework, individual life-courses are specified by sequences of state transitions (events) and the time spans between these transitions. The state space is usually discrete and finite, which is not necessarily the case in IBMPopSim, where individuals can have continuous characteristics. But most importantly, microsimulation does not allow for interactions between individuals. Indeed, microsimulation produces separately the life courses of all individuals in the populations, based on the computation of the distribution functions of the waiting times in the distinct states of the state space, for each individual (Zinn 2014). This can be slow in comparison to the simulation by thinning of event times occurring in the population, which is based on selecting event times among some competing proposed event times. Finally, MicSim simplifies the Mic-Core microsimulation tool implemented in Java (Zinn et al. 2009). However, the implementation in R of simulation algorithms yields longer simulation run times than when using Rcpp.

In Section 2, we give a short description of Stochastic Individual-Based Models (IBMs) and a quick example of model implementation with the IBMPopSim package. In Section 3, we introduce the mathematical framework that characterizes the class of IBMs that can be implemented in the IBMPopSim package. In particular, a general pathwise representation of IBMs is presented. The population dynamics is obtained as the solution of an SDE driven by Poisson measures, for which we obtain existence and uniqueness results in Theorem 3.1. In Section 4 the two main algorithms for simulating the population evolution of an IBM across the interval $[0, T]$ are detailed. In Section 5 we present the main functions of the IBMPopSim package, which allow for the definition of events and their intensities, the creation of a model, and the simulation of scenarios. Two examples are detailed in Section 6 and Section 7, featuring applications involving an heterogeneous insurance portfolio characterized by entry and exit events, and an age and size-structured population with intricate interactions.

## 2 Brief overview of IBMPopSim

Stochastic Individual-Based Models (IBMs) represent a broad class of random population dynamics models, allowing the description of population evolution on an individual scale. Informally, an IBM can be summarized by the description of the individuals constituting the population, the various types of events that can occur to these individuals, along with their respective frequencies. In IBMPopSim, individuals can be characterized by their age and/or a collection of discrete or continuous characteristics. Moreover, the package enables users to simulate efficiently populations in which one or more of the following event types may occur:

- **Birth event**: addition of an individual of age 0 to the population.
- **Death event**: removal of an individual from the population.
- **Entry event**: arrival of an individual in the population.
- **Exit (emigration) event**: exit from the population (other than death).
- **Swap event**: an individual changes characteristics.

Each event type is linked to an associated event kernel, describing how the population is modified following the occurrence of the event. For some event types, the event kernel requires explicit specification. This is the case for entry events when a new individual joins the population, which requires to specify the age and characteristics of this new individual. For instance, the characteristics of a new individual in the population can be chosen uniformly in the space of all characteristics, or can depend on the distribution of his parents or those of the other individuals composing the population.

The last component of an IBM are the event intensities. Informally, an event intensity is a function

$\lambda_t^e(I, Z)$ describing the frequency at which an event $e$ can occur to an individual $I$ in a population $Z$ at a time $t$. Given a history of the population $(\mathscr{F}_t)$, the probability of event $e$ occurring to individual $I$ during a small interval of time $(t, t + dt]$ is proportional to $\lambda^e(I, t)$:

$$\mathbb{P}(\text{event } e \text{ occurring to } I \text{ during } (t, t + dt] | \mathscr{F}_t) \simeq \lambda_t^e(I, Z)dt.$$

The intensity function $\lambda^e$ can include various dependencies:

- **individual intensity**: $\lambda^e$ depends only on the individual's $I$ age and characteristics, and time $t$,
- **interaction intensity**: in addition $\lambda^e$ depends on the population composition $Z$.

Prior to providing a detailed description of an Individual-Based Model (IBM), we present a simple model of birth and death in an age-structured *human* population. We assume no interactions between individuals, and individuals are characterized by their gender, in addition to their age. In this simple model, all individuals, regardless of gender, can give birth when their age falls between 15 and 40 years, with a constant birth rate of 0.05. The death intensity is assumed to follow a Gompertz-type intensity depending on age. The birth and death intensities are then given by

$$\lambda^b(t, I) = 0.05 \times \mathbf{1}_{[15,40]}(a(I, t)), \quad \lambda^d(t, I) = \alpha \exp(\beta a(I, t)),$$

with $a(I, t)$ the age of individual $I$ at time $t$. Birth events are also characterized with a kernel determining the gender of the newborn, who is male with probability $p_{male}$.

## 2.1 Model creation

All models in `IBMPopSim` are created with a call to the `mk_model` function, which takes the list of events as an argument. In this example, the events are created with the `mk_event_individual` function, involving a few lines of `cpp` instructions defining the intensity and, if applicable, the kernel of the event. For a more in depth description of the event creation step and its parameters, we refer to Section 5.2.

The events of this simple model are for example defined through the following calls.

```
birth_event <- mk_event_individual(
  type = "birth",
  intensity_code = "result = birth_rate(I.age(t));",
  kernel_code = "newI.male = CUnif(0,1) < p_male;")

death_event <- mk_event_individual(
  type = "death",
  intensity_code = "result = alpha * exp(beta * I.age(t));")
```

In the `cpp` codes, the names `birth_rate`, `p_male`, `alpha` and `beta` refer to the model parameters defined in the following list.

```
params <- list(
  "alpha" = 0.008, "beta" = 0.02,
  "p_male" = 0.51,
  "birth_rate" = stepfun(c(15, 40), c(0, 0.05, 0)))
```

In a second step, the model is created by calling the function `mk_model`. A `cpp` source code is automatically created through a template mechanism based on the events and parameters, subsequently compiled using the `sourceCpp` function from the `Rcpp` package.

```
birth_death_model <- mk_model(
  characteristics = c("male" = "bool"),
  events = list(death_event, birth_event),
  parameters = params)
```

## 2.2 Simulation

Once the model is created and compiled, the `popsim` function is called to simulate the evolution of a population according to this model. To achieve this, an initial population must be defined. In this example, we extract a population from a dataset specified in the package (a sample of 100 000 individuals based on the population of England and Wales in 2014). It is also necessary to set bounds for the events intensities. In this example, they are obtained by assuming that the maximum age for an individual is 115 years.

```
a_max <- 115
events_bounds = c(
  "death" = params$alpha * exp(params$beta * a_max),
  "birth" = max(params$birth_rate))
```

The function `popsim` can now be called to simulate the population starting from the initial population `population(EW_pop_14$sample)` up to time $T = 30$.

```
sim_out <- popsim(
  birth_death_model,
  population(EW_pop_14$sample),
  events_bounds,
  parameters = params, age_max = a_max,
  time = 30)
```

The data frame `sim_out$population` contains the information (birth, death, gender) on individuals who lived in the population over the period $[0, 30]$. Functions of the package allows to provide aggregated information on the population.

# 3 Mathematical framework

In this section, we define rigorously the class of IBMs that can be simulated in `IBMPopSim`, along with the assumptions that are required in order for the population to be simulatable. The representation of age-structured IBMs based on measure-valued processes, as introduced in Tran (2008), is generalized to a wider class of abstract population dynamics. The modeling differs slightly here, since individuals are *kept in the population* after their death (or exit), by including the death/exit date as an individual trait.

In the remainder of the paper, the filtered probability space is denoted by $(\Omega, \{\mathscr{F}_t\}, \mathbb{P})$, under the usual assumptions. All processes are assumed to be càdlàg and adapted to the filtration $\{\mathscr{F}_t\}$ (for instance the history of the population) on a time interval $[0, T]$. For a càdlàg process $X$, we denote $X_{t^-} := \lim_{\substack{s \to t \\ s < t}} X_s$.

## 3.1 Population

As mentioned in Section 2 a population is a collection of individuals whose evolution defines the population process.

### 3.1.1 Individuals

An individual is represented by a triplet $I = (\tau^b, \tau^d, x) \in \mathcal{I} = \mathbb{R} \times \bar{\mathbb{R}} \times \mathcal{X}$ with:

- $\tau^b \in \mathbb{R}$ the date of birth,
- $\tau^d \in \bar{\mathbb{R}}$ the death date, with $\tau^d = \infty$ if the individual is still alive,
- a collection $x \in \mathcal{X}$ of characteristics where $\mathcal{X}$ is the space of characteristics.

Note that in IBMs, individuals are usually characterized by their age $a(t) = t - \tau^b$ instead of their date of birth $\tau^b$. However, using the latter is actually easier for the simulation, as it remains constant over time.

### 3.1.2 Population process

The population at a given time $t$ is a random set

$$Z_t = \{I_k \in \mathcal{I}; \ k = 1, \dots, N_t\},$$

composed of all individuals (alive or dead) who have lived in the population before time $t$. As a random set, $Z_t$ can be represented by a random counting measure on $\mathcal{I}$, that is an integer-valued measure $Z : \Omega \times \mathcal{I} \to \bar{\mathbb{N}}$ where for $A \in \mathcal{I}$, $Z(A)$ is the (random) number of individuals $I$ in the subset $A$. With this representation:

$$
\begin{aligned}
Z_t(\mathrm{d}\tau^b, \mathrm{d}\tau^d, \mathrm{d}x) &= \sum_{k=1}^{N_t} \delta_{I_k}(\tau^b, \tau^d, x), \\
\text{with } \int_{\mathcal{I}} f(\tau^b, \tau^d, x) Z_t(\mathrm{d}\tau^b, \mathrm{d}\tau^d, \mathrm{d}x) &= \sum_{k=1}^{N_t} f(I_k).
\end{aligned}
\tag{1}
$$

The number of individuals present in the population *before time $t$* is obtained by taking $f \equiv 1$:

$$N_t = \int_{\mathcal{I}} Z_t(\mathrm{d}\tau^b, \mathrm{d}\tau^d, \mathrm{d}x) = \sum_{k=1}^{N_t} \mathbf{1}_{\mathcal{I}}(I_k).$$

Note that $(N_t)_{t \geq 0}$ is an increasing process since dead/exited individuals are kept in the population $Z$. The number of alive individuals in the population at time $t$ is:

$$N_t^a = \int_{\mathcal{I}} \mathbf{1}_{\{\tau^d > t\}} Z_t(\mathrm{d}\tau^b, \mathrm{d}\tau^d, \mathrm{d}x) = \sum_{k=1}^{N_t} \mathbf{1}_{\{\tau_k^d > t\}}.
\tag{2}$$

Another example is the number of alive individuals of age over $a$ is

$$N_t([a, +\infty)) := \int_{\mathcal{I}} \mathbf{1}_{[a, +\infty)}(t - \tau^b) \mathbf{1}_{]t, \infty]}(\tau^d) Z_t(\mathrm{d}\tau^b, \mathrm{d}\tau^d, \mathrm{d}x) = \sum_{k=1}^{N_t} \mathbf{1}_{\{t - \tau_k^b \geq a\}} \mathbf{1}_{\{\tau_k^d \geq t\}}.$$

## 3.2 Events

The population composition changes at random dates following different types of events. `IBMPopSim` allows the simulation of IBMs with the following events types:

- A **birth** event at time $t$ is the addition of a new individual $I' = (t, \infty, X)$ of age 0 to the population. Their date of birth is $\tau^b = t$, and characteristics is $X$, a random variable of distribution defined by the birth kernel $k^b(t, I, \mathrm{d}x)$ on $\mathcal{X}$, depending on $t$ and its parent $I$. The population size becomes $N_t = N_{t^-} + 1$, and the population composition after the event is

$$Z_t = Z_{t^-} + \delta_{(t, \infty, X)}.$$

- An **entry** event at time $t$ is also the addition of an individual $I'$ in the population. However, this individual is not of age 0. The date of birth and characteristics of the new individual $I' = (\tau^b, \infty, X)$ are random variables of probability distribution defined by the entry kernel $k^{en}(t, ds, dx)$ on $\mathbb{R} \times \mathcal{X}$. The population size becomes $N_t = N_{t^-} + 1$, and the population composition after the event is:

$$Z_t = Z_{t^-} + \delta_{(\tau^b, \infty, X)}.$$

- A **death** or **exit** event of an individual $I = (\tau^b, \infty, x) \in Z_{t^-}$ at time $t$ is the modification of its death date $\tau^d$ from $+\infty$ to $t$. This event results in the simultaneous addition of the individual $(\tau^b, t, x)$ and removal of the individual $I$ from the population. The population size is not modified, and the population composition after the event is

$$Z_t = Z_{t^-} + \delta_{(\tau^b, t, x)} - \delta_I.$$

- A **swap** event (change of characteristics) results in the simultaneous addition and removal of an individual. If an individual $I = (\tau^b, \infty, x) \in Z_{t^-}$ changes of characteristics at time $t$, then it is removed from the population and replaced by $I' = (\tau^b, \infty, X)$. The new characteristics $X$ is a random variable of distribution $k^s(t, I, dx)$ on $\mathcal{X}$, depending on time, the individual's age and previous characteristics $x$. In this case, the population size is not modified and the population becomes:

$$Z_t = Z_{t^-} + \delta_{(\tau^b, \infty, X)} - \delta_{(\tau^b, \infty, x)}.$$

To summarize, the space of event types is $E = \{b, en, d, s\}$, and the jump $\Delta Z_t = Z_t - Z_{t^-}$ (change in the population composition) generated by an event of type $e \in \{b, en, d, s\}$ is denoted by $\phi^e(t, I)$. We thus have the following rules summarized in Table 1.

Table 1: Action in the population for a given event type

| Event | Type | $\phi^e(t, I)$ | New individual |
|---|---|---|---|
| Birth | $b$ | $\delta_{(t, \infty, X)}$ | $\tau^b = t,\ X \sim k^b(t, I, dx)$ |
| Entry | $en$ | $\delta_{(\tau^b, \infty, X)}$ | $(\tau^b, X) \sim k^{en}(t, ds, dx)$ |
| Death/Exit | $d$ | $\delta_{(\tau^b, t, x)} - \delta_{(\tau^b, \infty, x)}$ | $\tau^d = t$ |
| Swap | $s$ | $\delta_{(\tau^b, \infty, X)} - \delta_{(\tau^b, \infty, x)}$ | $X \sim k^s(t, I, dx)$ |

*Remark* 3.1 (Composition of the population).

- At time $T$, the population $Z_T$ contains all individuals who lived in the population before $T$, including dead/exited individuals. If there are no swap events, or entries, the population state $Z_t$ for any time $t \leq T$ can be obtained from $Z_T$. Indeed, if $Z_T = \sum_{k=1}^{N_T} \delta_{I_k}$, then the population at time $t \leq T$ is simply composed of the individuals born before $t$:

$$Z_t = \sum_{k=1}^{N_T} \mathbf{1}_{\{\tau_k^b \leq t\}} \delta_{I_k}.$$

- In the presence of entries (open population), a characteristic $x$ can track the individuals' entry dates. Then, the previous equation can be easily modified in order to obtain the population $Z_t$ at time $t \leq T$ from $Z_T$.

## 3.3 Events intensity

Once the different event types have been defined in the population model, the frequency at which each event $e$ occurs in the population has to be specified. Informally, the intensity $\Lambda_t^e(Z_t)$ at which an event $e$ can occur is defined by

$$\mathbb{P}\big(\text{event } e \text{ occurs in the population } Z_t \in (t, t+\mathrm{d}t]|\mathscr{F}_t\big) \simeq \Lambda_t^e(Z_t)\mathrm{d}t.$$

For a more formal definition of stochastic intensities, we refer to Brémaud (1981) or Kaakai and El Karoui (2023). The form of the intensity function $(\Lambda_t^e(Z_t))$ determines the population simulation algorithm in IBMPopSim:

- When the event intensity does not depend on the population state,

$$\big(\Lambda_t^e(Z_t)\big)_{t\in[0,T]} = \big(\mu^e(t)\big)_{t\in[0,T]}, \tag{3}$$

  with $\mu^e$ a deterministic function, the events of type $e$ occur at the jump times of an inhomogeneous Poisson process of intensity function $(\mu^e(t))_{t\in[0,T]}$. This is particularly useful when entry events occur with intensities influenced by environmental processes and/or exhibit seasonal variations. When such an event occurs, the individual to whom the event happens is drawn uniformly from the living individuals in the population. In a given model, the set of events $e \in E$ with Poisson intensities will be denoted by $\mathscr{P}$.

- Otherwise, we assume that the global intensity $\Lambda_t^e(Z_t)$ at which the events of type $e$ occur in the population can be written as the sum of individual intensities $\lambda_t^e(I, Z_t)$:

$$\Lambda_t^e(Z_t) = \sum_{k=1}^{N_t} \lambda_t^e(I_k, Z_t), \tag{4}$$

  with $\mathbb{P}\big(\text{event } e \text{ occurs to an individual } I \in (t, t+\mathrm{d}t]|\mathscr{F}_t\big) \simeq \lambda_t^e(I, Z_t)\mathrm{d}t.$

Obviously, nothing can happen to dead or exited individuals, i.e. individuals $I = (\tau^b, \tau^d, x)$ with $\tau^d \leq t$. Thus, individual event intensities are assumed to be null for dead/exited individuals:

$$\lambda_t^e(I, Z_t) = 0, \text{ if } \tau^d \leq t, \text{ so that } \Lambda_t^e(Z_t) = \sum_{k=1}^{N_t^a} \lambda_t^e(I_k, Z_t),$$

with $N_t^a$ the number of alive individuals at time $t$.

The event's individual intensity $\lambda_t^e(I, Z_t)$ can depend on time (for instance when there is a mortality reduction over time), on the individual's age $t - \tau^b$ and characteristics, but also on the population composition $Z_t$. The dependence of $\lambda^e$ on the population $Z$ models interactions between individuals in the populations. Hence, two types of individual intensity functions can be implemented in IBMPopSim:

1. *No interactions:* The intensity function $\lambda^e$ does not depend on the population composition. The intensity at which an event of type $e$ occurs to an individual $I$ only depends on its date of birth and characteristics:

$$\lambda_t^e(I, Z_t) = \lambda^e(t, I), \tag{5}$$

   where $\lambda^e : \mathbb{R}_+ \times \mathscr{I} \to \mathbb{R}^+$ is a deterministic function. In a given model, we denote by $\mathscr{E}$ the set of event types with individual intensity Equation 5.

2. *"Quadratic" interactions:* The intensity at which an event of type $e$ occurs to an individual $I$ depends on $I$ and on the population composition, through an interaction function $W^e$. The

quantity $W^e(t, I, J)$ describes the intensity of interactions between two alive individuals $I$ and $J$ at time $t$, for instance in the presence of competition or cooperation. In this case, we have

$$\lambda_t^e(I, Z_t) = \sum_{j=1}^{N_t} W^e(t, I, I_j) = \int_{\mathcal{I}} W^e(t, I, (\tau^b, \tau^d, x)) Z_t(\mathrm{d}\tau^b, \mathrm{d}\tau^d, \mathrm{d}x), \tag{6}$$

where $W^e(t, I, (\tau^b, \tau^d, x)) = 0$ if the individual $J = (\tau^b, \tau^d, x)$ is dead, i.e. $\tau^d \leq t$. In a given model, we denote by $\mathcal{E}_W$ the set of event types with individual intensity Equation 6.

To summarize, an individual intensity in IBMPopSim can be written as:

$$\lambda_t^e(I, Z_t) = \lambda^e(t, I)\mathbf{1}_{\{e \in \mathcal{E}\}} + \left(\sum_{j=1}^{N_t} W^e(t, I, I_j)\right)\mathbf{1}_{\{e \in \mathcal{E}_W\}}. \tag{7}$$

**Example 3.1.**

1. An example of death intensity without interaction for an individual $I = (\tau^b, \tau^d, x)$ alive at time $t$, $t < \tau^d$, is:
$$\lambda^d(t, I) = \alpha_x \exp(\beta_x a(I, t)), \text{ where } a(I, t) = t - \tau^b$$
is the age of the individual $I$ at time $t$. In this standard case, the death rate of an individual $I$ is an exponential (Gompertz) function of the individual's age, with coefficients depending on the individual's characteristics $x$.

2. In the presence of competition between individuals, the death intensity of an individual $I$ also depends on other individuals $J$ in the population. For example, if $I = (\tau^b, \tau^d, x)$, with its size $x$, then we have:
$$W^d(t, I, J) = (x_J - x)^+ \mathbf{1}_{\{\tau_J^d > t\}}, \quad \forall J = (\tau_J^b, \tau_J^d, x_J). \tag{8}$$
This can be interpreted as follows: if the individual $I$ meets randomly an individual $J$ alive at time $t$, and of bigger size $x_J > x$, then he can die at the intensity $x_J - x$. If $J$ is smaller than $I$, then it cannot kill $I$. The bigger is the size $x$ of $I$, the lower is its death intensity $\lambda_t^d(I, Z_t)$ defined by
$$\lambda_t^d(I, Z_t) = \sum_{\substack{J \in Z_t, \\ x_J > x}} (x_J - x)\mathbf{1}_{\{\tau_J^d > t\}}.$$

3. `IBMPopSim` can simulate IBMs that include intensities expressed as a sum of Poisson intensities and individual intensities of the form $\Lambda^e(Z_t) = \mu_t^e + \sum_{k=1}^{N_t} \lambda^e(I_k, Z_t)$. Other examples are provided in Section 6 and Section 7.

Finally, the global intensity at which an event can occur in the population is defined by:

$$\Lambda_t(Z_t) = \sum_{e \in \mathcal{P}} \mu^e(t) + \sum_{e \in \mathcal{E}} \left(\sum_{k=1}^{N_t} \lambda^e(t, I_k)\right) + \sum_{e \in \mathcal{E}_W} \left(\sum_{k=1}^{N_t} \sum_{j=1}^{N_t} W^e(t, I_k, I_j)\right). \tag{9}$$

An important point is that for events $e \in \mathcal{E}$ without interactions, the global event intensity $\Lambda_t^e(Z_t) = \sum_{k=1}^{N_t} \lambda^e(t, I_k)$ is *of order* $N_t^a$ defined in Equation 2 (number of alive individuals at time $t$). On the other hand, for events $e \in \mathcal{E}_W$ with interactions, $\Lambda_t^e(Z_t) = \sum_{k=1}^{N_t} \sum_{j=1}^{N_t} W^e(t, I_k, I_j)$ is of order $(N_t^a)^2$. Informally, this means that when the population size increases, events with interaction are more costly to simulate. Furthermore, the numerous computations of the interaction kernel $W^e$ can also be computationally costly. The randomized Algorithm 3 , detailed in Section 4.3, allows us to overcome these limitations.

**Events intensity bounds**

The simulation algorithms implemented in `IBMPopSim` are based on an acceptance/rejection procedure, which requires the user to specify bounds for the various events intensities $\Lambda_t^e(Z_t)$. These bounds are defined differently depending on the expression of the intensity.

**Assumption 3.1.** *For all events $e \in \mathscr{P}$ with Poisson intensity (Equation 3), the intensity is assumed to be bounded on $[0, T]$:*

$$\forall t \in [0, T], \quad \Lambda_t^e(Z_t) = \mu^e(t) \leq \bar{\mu}^e.$$

When $e \in \mathscr{E} \cup \mathscr{E}_W$, $\Lambda_t^e(Z_t) = \sum_{k=1}^{N_t} \lambda_t^e(I_k, Z_t)$, assuming that $\Lambda_t^e(Z_t)$ is uniformly bounded is too restrictive since the event intensity depends on the population size. In this case, the assumption is made on the individual intensity $\lambda^e$ or on the interaction function $W^e$, depending on the situation.

**Assumption 3.2.** *For all event types $e \in \mathscr{E}$, the associated individual event intensity $\lambda^e$ with no interactions (Equation 5) is assumed to be uniformly bounded:*

$$\lambda^e(t, I) \leq \bar{\lambda}^e, \quad \forall\, t \in [0, T],\ I \in \mathscr{I}.$$

*In particular,*

$$\forall t \in [0, T], \quad \Lambda_t^e(Z_t) = \sum_{k=1}^{N_t} \lambda^e(t, I) \leq \bar{\lambda}^e N_t. \tag{10}$$

**Assumption 3.3.** *For all event types $e \in \mathscr{E}_W$, the associated interaction function $W^e$ is assumed to be uniformly bounded:*

$$W^e(t, I, J) \leq \bar{W}^e, \quad \forall\, t \in [0, T],\ I, J \in \mathscr{I}.$$

*In particular, $\forall t \in [0, T]$,*

$$\lambda_t^e(I, Z_t) = \sum_{j=1}^{N_t} W^e(t, I, I_j) \leq \bar{W}^e N_t, \quad and \quad \Lambda_t^e(Z_t) \leq \bar{W}^e(N_t)^2.$$

Assumption 3.1, Assumption 3.2 and Assumption 3.3 yield that events in the population occur with the global event intensity $\Lambda_t(Z_t)$, given in Equation 9, which is dominated by a polynomial function in the population size:

$$\Lambda_t(Z_t) \leq \bar{\Lambda}(N_t), \quad \text{with } \bar{\Lambda}(n) = \sum_{e \in \mathscr{P}} \bar{\mu}^e + \sum_{e \in \mathscr{E}} \bar{\lambda}^e n + \sum_{e \in \mathscr{E}_W} \bar{W}^e n^2. \tag{11}$$

This bound is linear in the population size if there are no interactions, and quadratic if there at least is an event including interactions. This assumption is the key to the algorithms implemented in `IBMPopSim`. Before presenting the simulation algorithm, we close this section with a rigorous definition of an IBM, based on the pathwise representation of its dynamics as a Stochastic Differential Equation (SDE) driven by Poisson random measures.

## 3.4 Pathwise representation of stochastic IBM

Since the seminal paper of Fournier and Méléard (2004), it has been shown in many examples that a stochastic IBM dynamics can be defined rigorously as the unique solution of an SDE driven by Poisson measures, under reasonable non explosion conditions. In the following, we introduce a unified framework for the pathwise representation of the class of stochastic IBMs introduced above. Some recalls on Poisson random measures are presented in the Appendix Section 8.1, and for more details on these representations of particular examples, we refer to the abundant literature on the subject (see Çinlar (2011) and the references therein).

In the following we consider an individual-based stochastic population $(Z_t)_{t \in [0,T]}$, keeping the notations introduced in Section 3.2 and Section 3.3 for the events and their intensities. In particular, the set of events types that define the population evolution is denoted by $\mathscr{P} \cup \mathscr{E} \cup \mathscr{E}_W \subset E$, with $\mathscr{P}$ the set of events types with Poisson intensity verifying Assumption 3.1, $\mathscr{E}$ the set of events types with individual intensity and no interaction, verifying Assumption 3.2 and finally $\mathscr{E}_W$ the set of event types with interactions, verifying Assumption 3.3.

**Non-explosion criterion**

First, one has to ensure that the number of events occurring in the population will not explode in finite time, leading to an infinite simulation time. Assumption 3.2 and Assumption 3.3 are not sufficient to guarantee the non explosion of the event number, due to the potential explosion of the population size in the presence of interactions. An example is the case when only birth events occur, with an intensity $\Lambda_t^b(Z_t) = C_b(N_t^a)^2$ (i.e. when $W^b(t, I, J) = C_b$). Then, the number of alive individuals $(N_t^a)_{t \geq 0}$ is a well-known pure birth process of intensity function $g(n) = C_b n^2$ (intensity of moving from state $n$ to $n + 1$). This process explodes in finite time, since $g$ does not verify the necessary and sufficient non explosion criterion for pure birth Markov processes: $\sum_{n=1}^{\infty} \frac{1}{g(n)} = \infty$ (see e.g. Theorem 2.2 in (Bansaye and Méléard 2015)). There is thus an explosion in finite time of birth events.

This example shows that the important point for non explosion is to control the population size. We give below a general sufficient condition on birth and entry event intensities, in order for the population size to stay finite in finite time. This ensures that the number of events does not explode in finite time. Informally, the idea is to control the intensities by a pure birth intensity function verifying the non-explosion criterion.

**Assumption 3.4.** *Let $e \in \{b, en\}$ a birth or entry event type. If the intensity at which the events of type $e$ occur in the population are not Poissonian, i.e. $e \in \mathscr{E} \cup \mathscr{E}_W$, then there exists a function $f^e : \mathbb{N} \to (0, +\infty)$, such that*

$$\sum_{n=1}^{\infty} \frac{1}{n f^e(n)} = \infty,$$

*and for all individual $I \in \mathscr{I}$ and population measure $Z = \sum_{k=1}^{n} \delta_{I_k}$ of size $n$,*

$$\lambda_t^e(I, Z) \leq f^e(n), \ \forall \ 0 \leq t \leq T.$$

If $e \in \mathscr{E}$, $\lambda_t^e(I, Z) = \lambda^e(t, I) \leq \bar{\lambda}^e$ by the domination Assumption 3.3, then Assumption 3.4 is always verified with $f^e(n) = \bar{\lambda}^e$.

Assumption 3.4 yields that the global intensity $\Lambda_t^e(\cdot)$ of event $e$ is bounded by a function $g^e$ only depending on the population size:

$$\Lambda_t^e(Z) \leq g^e(n) := n f^e(n), \quad \text{with } \sum_{n=1}^{\infty} \frac{1}{g^e(n)} = \infty.$$

If $e \in \mathscr{P}$ has a Poisson intensity, then $\Lambda_t^e(Z) = \mu_t^e$ always verifies the previous equation with $g^e(n) = \bar{\mu}^e$.

Before introducing the IBM SDE, let us give an idea of the equation construction. Between two successive events, the population composition $Z_t$ stays constant, since the population process $(Z_t)_{t \geq 0}$ is a pure jump process. Furthermore, since each event type is characterized by an intensity function, the jumps occurring in the population can be represented by restriction and projection of a Poisson measure defined on a larger state space. More precisely, we introduce a random Poisson measure $Q$ on $\mathbb{R}^+ \times \mathscr{J} \times \mathbb{R}^+$, with $\mathscr{J} = \mathbb{N} \times (\mathscr{E} \cup \mathscr{E}_W)$. $Q$ is composed of random quadruplets $(\tau, k, e, \theta)$, where $\tau$ represents a potential event time for an individual $I_k$ and event type $e$. The last variable $\theta$ is used to

accept/reject this proposed event, depending on the event intensity. Hence, the Poisson measure is restricted to a certain random set and then projected on the space of interest $\mathbb{R}^+ \times \mathcal{J}$. If the event is accepted, then a jump $\phi^e(\tau, I_k)$ occurs.

**Theorem 3.1** (Pathwise representation). *Let $T \in \mathbb{R}^+$ and $\mathcal{J} = \mathbb{N} \times (\mathcal{E} \cup \mathcal{E}_W)$. Let $Q$ be a random Poisson measure on $\mathbb{R}^+ \times \mathcal{J} \times \mathbb{R}^+$, of intensity $\mathrm{d}t \delta_{\mathcal{J}}(\mathrm{d}k, \mathrm{d}e)(\theta)\mathrm{d}\theta$, with $\delta_{\mathcal{J}}$ the counting measure on $\mathcal{J}$. Finally, let $Q^{\mathcal{P}}$ be a random Poisson measure on $\mathbb{R}^+ \times \mathcal{P} \times \mathbb{R}^+$, of intensity $\mathrm{d}t \delta_P(\mathrm{d}e)\mathrm{d}\theta$, and $Z_0 = \sum_{k=1}^{N_0} \delta_{I_k}$ an initial population. Then, under Assumption 3.4 , there exists a unique measure-valued population process $Z$, strong solution on the following SDE driven by the Poisson measure $Q$:*

$$
\begin{aligned}
Z_t = Z_0 &+ \int_0^t \int_{\mathcal{J} \times \mathbb{R}^+} \phi^e(s, I_k) \mathbf{1}_{\{k \leq N_{s^-}\}} \mathbf{1}_{\{\theta \leq \lambda_s^e(I_k, Z_{s^-})\}} Q(\mathrm{d}s, \mathrm{d}k, \mathrm{d}e, \mathrm{d}\theta) \\
&+ \int_0^t \int_{\mathcal{P} \times \mathbb{R}^+} \phi^e(s, I_{s^-}) \mathbf{1}_{\{\theta \leq \mu^e(s)\}} Q^{\mathcal{P}}(\mathrm{d}s, \mathrm{d}e, \mathrm{d}\theta), \qquad \forall 0 \leq t \leq T,
\end{aligned}
\tag{12}
$$

*and where $I_{s^-}$ is an individual, chosen uniformly among alive individuals in the population $Z_{s^-}$.*

The proof of Theorem 3.1 is detailed in the Appendix, Section 8.2.1. Note that Equation 12 is an SDE describing the evolution of the IBM, the intensity of the events in the right hand side of the equation depending on the population process $Z$ itself. The main idea of the proof of Theorem 3.1 is to use the non explosion property of Lemma 3.1, and to write the r.h.s of Equation 12 as a sum of simple equations between two successive events, solved by induction.

**Lemma 3.1.** *Let $Z$ be a solution of Equation 12 on $\mathbb{R}^+$, with $(T_n)_{n \geq 0}$ its jump times, $T_0 = 0$. If Assumption 3.4 is satisfied, then*

$$
\lim_{n \to \infty} T_n = \infty, \quad \mathbb{P}\text{-a.s.}
$$

The proof of Lemma 3.1, detailed in Appendix Section 8.2.2 is more technical and relies on a pathwise comparison result, generalizing those obtained in (Kaakai and El Karoui 2023). An alternative pathwise representation of the population process, inspired by the randomized Algorithm 3 is given as well in Proposition 4.3.

# 4 Population simulation

We now present the main algorithm for simulating the evolution of an IBM over $[0, T]$. The algorithm implemented in `IBMPopSim` allows the exact simulation of Equation 12, based on an acceptance/reject algorithm for simulating random times called *thinning*. The exact simulation of event times with this acceptance/reject procedure is closely related to the simulations of inhomogeneous Poisson processes by the so-called thinning algorithm, often attributed to Lewis and Shedler (1979). The simulation methods for inhomogeneous Poisson processes can be adapted to IBMs, and we introduce in this section a general algorithm extending those by Fournier and Méléard (2004) (see also Ferrière and Tran (2009), Bensusan (2010)).

It can be noted that under appropriate rescaling and when the population size goes to infinity, an IBM can be approximated by a non linear transport PDE, structured by age and trait. A central limit theorem can also be obtained under appropriate assumptions (Tran 2008). In the presence of interactions as in Section 7 for instance, the IBM goes almost surely to extinction in finite time, which is not the case for the limit PDE. In this case, simulating the microscopic process can be quite useful for approximating the distribution of the extinction time. Other applications of IBM simulations can include the simulation of multiscale population evolution, strongly heterogeneous populations, or small populations with strong interactions.

The algorithm is based on exponential "candidate" event times, chosen with a (constant) intensity which must be greater than the global event intensity $\Lambda_t(Z_t)$ (Equation 4). Starting from time $t$, once a candidate event time $t + \bar{T}_\ell$ has been proposed, a candidate event type $e$ (birth, death,...) is chosen with a probability $p^e$ depending on the event intensity bounds $\bar{\mu}^e$, $\bar{\lambda}^e$ and $\bar{W}^e$, as defined in Assumption 3.2 and Assumption 3.3. An individual $I$ is then drawn from the population. Finally, it remains to accept or reject the candidate event with a probability $q^e(t, I, Z_t)$ depending on the true event intensity. If the candidate event time is accepted, then the event $e$ occurs at time $t + \bar{T}_\ell$ to the individual $I$. The main idea of the implemented algorithm can be summarized as follows:

1. Draw a candidate time $t + \bar{T}_\ell$ and candidate event type $e$.
2. Draw a uniform variable $\theta \sim \mathcal{U}([0, 1])$ and individual $I$.
3. **If** $\theta \leq q^e(t, I, Z_t)$ **then** event $e$ occur to individual $I$, **else** Do nothing and start again from $t + \bar{T}_\ell$.

Before introducing the main algorithms in more details, we recall briefly the thinning procedure for simulating inhomogeneous Poisson processes, as well as the links with pathwise representations. Some recalls on Poisson random measures are presented in Section 8.1. For a more general presentation of thinning of a Poisson random measure, see (Devroye 1986; Çinlar 2011; Kallenberg 2017).

## 4.1 Thinning of Poisson measure

Let us start with the simulation and pathwise representation of an inhomogeneous Poisson process on $[0, T]$ with intensity $(\Lambda(t))_{t \in [0,T]}$. The thinning procedure is based on the fundamental assumption that $\Lambda(t) \leq \bar{\Lambda}$ is bounded on $[0, T]$. In this case, the inhomogeneous Poisson can be obtained from an homogeneous Poisson process of intensity $\bar{\Lambda}$, which can be simulated easily.

First, the Poisson process can be extended to a Marked Poisson measure $\bar{Q} := \sum_{\ell \geq 1} \delta_{(\bar{T}_\ell, \bar{\Theta}_\ell)}$ on $(\mathbb{R}^+)^2$, defined as follow:

- The jump times of $(\bar{T}_\ell)_{\ell \geq 1}$ of $\bar{Q}$ are the jump times of a Poisson process of intensity $\bar{\Lambda}$.

- The marks $(\bar{\Theta}_\ell)_{\ell \geq 1}$ are *i.i.d.* random variables, uniformly distributed on $[0, \bar{\Lambda}]$.

By Proposition 8.3, $\bar{Q}$ is a Poisson random measure with mean measure

$$\bar{\mu}(\mathrm{d}t, \mathrm{d}\theta) := \bar{\Lambda}\mathrm{d}t\frac{\mathbf{1}_{[0,\bar{\Lambda}]}(\theta)}{\bar{\Lambda}}\mathrm{d}\theta = \mathrm{d}t\mathbf{1}_{[0,\bar{\Lambda}]}(\theta)\mathrm{d}\theta.$$

In particular, the average number of atoms $(\bar{T}_\ell, \bar{\Theta}_\ell)$ in $[0, t] \times [0, h]$ is

$$\mathbb{E}[Q([0,t] \times [0,h])] = \mathbb{E}[\sum_\ell \mathbf{1}_{[0,t]\times[0,h]}(\bar{T}_\ell, \bar{\Theta}_\ell)] = \int_{(\mathbb{R}^+)^2} \bar{\mu}(\mathrm{d}t, \mathrm{d}\theta) = t(\bar{\Lambda} \wedge h).$$

The thinning is based on the restriction property for Poisson measure: for a measurable set $\Delta \subset \mathbb{R}^+ \times \mathbb{R}^+$, the restriction $Q^\Delta := \mathbf{1}_\Delta \bar{Q}$ of $\bar{Q}$ to $\Delta$ (by taking only atoms in $\Delta$) is also a Poisson random measure of mean measure $\mu^\Delta(\mathrm{d}t, \mathrm{d}\theta) = \mathbf{1}_\Delta(t, \theta)\bar{\mu}(\mathrm{d}t, \mathrm{d}\theta)$.

In order to obtain an inhomogeneous Poisson measure of intensity $(\Lambda(t))$, the "good" choice of $\Delta$ is the hypograph of $\Lambda$: $\Delta = \{(t, \theta) \in [0, T] \times [0, \bar{\Lambda}]; \ \theta \leq \Lambda(t)\}$ (see Figure 1). Then,

$$Q^\Delta = \sum_{\ell \geq 1} \mathbf{1}_{\{\bar{\Theta}_\ell \leq \Lambda(\bar{T}_\ell)\}}\delta_{(\bar{T}_\ell, \bar{\Theta}_\ell)},$$

and since $\Lambda(t) \leq \bar{\Lambda}$, on $[0, T]$:

$$\mu^\Delta(\mathrm{d}t, \mathrm{d}\theta) = \mathbf{1}_{\{\theta \leq \Lambda(t)\}}\mathrm{d}t\mathbf{1}_{[0,\bar{\Lambda}]}(\theta)\mathrm{d}\theta = \mathbf{1}_{\{\theta \leq \Lambda(t)\}}\mathrm{d}t\mathrm{d}\theta.$$

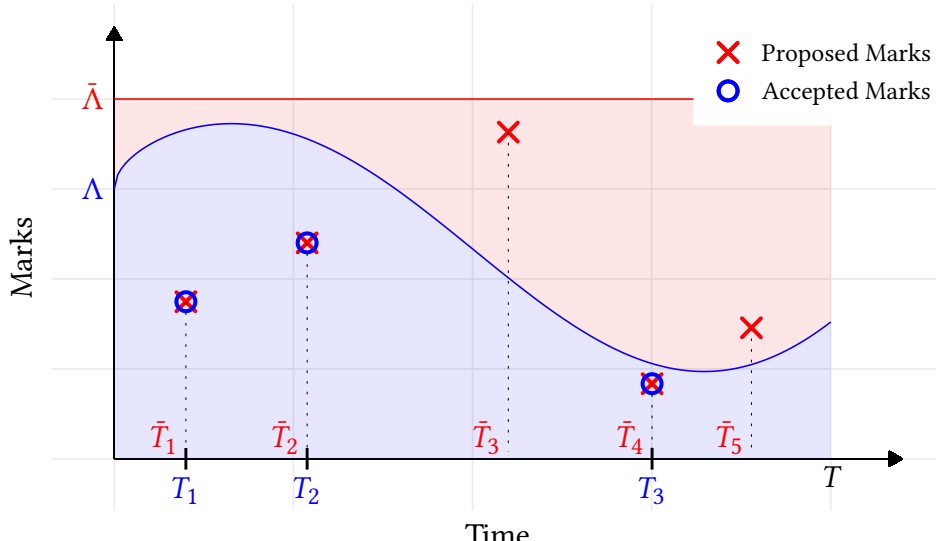

Figure 1: Realization of a Marked Poisson measure $\bar{Q}$ on $[0,T]$ with mean measure $\bar{\mu}(\mathrm{d}t, \mathrm{d}\theta) = \mathrm{d}t\mathbf{1}_{[0,\bar{\Lambda}]}(\theta)\mathrm{d}\theta$ (red crosses), and realization of the restriction $\bar{Q}^\Delta$ where $\Delta = \{(t,\theta) \in [0,T] \times [0,\bar{\Lambda}], \theta \leq \Lambda(t)\}$ (blue circles). The projection of $\bar{Q}^\Delta$ on first component is an inhomogeneous Poisson process on $[0,T]$ of intensity $(\Lambda(t))$ and jump times $(T_k)_{k\geq 1}$.

Finally, the inhomogeneous Poisson process is obtained by the projection Proposition 8.2, which states that the jump times of $Q^\Delta$ are the jump times of an inhomogeneous Poisson process of intensity $(\Lambda(t))$.

**Proposition 4.1.** *The counting process* $N^\Lambda$, *projection of* $Q^\Delta$ *on the time component and defined by,*

$$N_t^\Lambda := Q^\Delta([0,t] \times \mathbb{R}^+) = \int_0^t \int_{\mathbb{R}^+} \mathbf{1}_{\{\theta \leq \Lambda(s)\}} \bar{Q}(\mathrm{d}s, \mathrm{d}\theta) = \sum_{\ell \geq 1} \mathbf{1}_{\{\bar{T}_\ell \leq t\}} \mathbf{1}_{\{\bar{\Theta}_\ell \leq \Lambda(\bar{T}_\ell)\}}, \quad \forall t \in [0,T], \qquad (13)$$

*is an inhomogeneous Poisson process on* $[0,T]$ *of intensity function* $(\Lambda(t))_{t\in[0,T]}$. *The thinning Equation 13 is a pathwise representation of* $N^\Lambda$ *by restriction and projection of the Poisson measure* $Q$ *on* $[0,T]$.

The previous proposition yields a straightforward thinning algorithm to simulate the jump times $(T_k)_{k\geq 1}$ of an inhomogeneous Poisson process of intensity $\Lambda(t)$, by selecting jump times $\bar{T}_\ell$ such that $\bar{\Theta}_\ell \leq \Lambda(\bar{T}_\ell)$.

### 4.1.1 Multivariate Poisson process

This can be extended to the simulation of multivariate inhomogeneous Poisson processes, which is an important example before tackling the simulation of an IBM.

Let $(N^j)_{j \in \mathscr{J}}$ be a (inhomogeneous) multivariate Poisson process indexed by a finite set $\mathscr{J}$, such that $\forall j \in \mathscr{J}$, the intensity $(\lambda_j(t))_{t\in[0,T]}$ of $N_j$ is bounded on $[0,T]$:

$$\sup_{t\in[0,T]} \lambda_j(t) \leq \bar{\lambda}_j, \text{ and let } \bar{\Lambda} = \sum_{j \in \mathscr{J}} \bar{\lambda}_j.$$

Recall that such multivariate counting process can be rewritten as a Poisson random measure $N = \sum_{k \geq 1} \delta_{(T_k, J_k)}$ on $\mathbb{R}^+ \times \mathscr{J}$ (see e.g. Sec. 2 of Chapter 6 in (Çinlar 2011)), where $T_k$ is $k$th jump time of $\sum_{j\in\mathscr{J}} N^j$ and $J_k$ corresponds to the component of the the vector which jumps. In particular, $N_t^j = N([0,t] \times \{j\})$.

Once again the simulation of such process can be obtained from the simulation of a (homogeneous) multivariate Poisson process of intensity vector $(\bar{\lambda}_j)_{j \in \mathcal{J}}$, extended into a Poisson measures by adding marks on $\mathbb{R}^+$. Thus, we introduce the Marked Poisson measure $\bar{Q} = \sum \delta_{(\bar{T}_\ell, \bar{J}_\ell, \bar{\Theta}_\ell)}$ on $\mathbb{R}^+ \times \mathcal{J} \times \mathbb{R}^+$, such that:

- The jump times $(\bar{T}_\ell)$ of $\bar{Q}$ are the jump times of a Poisson measure of intensity $\bar{\Lambda}$.
- The variables $(\bar{J}_\ell)$ are *i.i.d.* random variables on $\mathcal{J}$, with $p_j = \mathbb{P}(\bar{J}_1 = j) = \bar{\lambda}_j / \bar{\Lambda}$ and representing the component of the vector which jumps.
- The marks $(\bar{\Theta}_\ell)$ are independent variables with $\bar{\Theta}_\ell$ a uniform random variable on $[0, \bar{\lambda}_{\bar{J}_\ell}]$, $\forall \ell \geq 1$.

By Proposition 8.3 and Proposition 8.2, each measure $\bar{Q}_j(\mathrm{d}t, \mathrm{d}\theta) = \bar{Q}(\mathrm{d}t, \{j\}, \mathrm{d}\theta) = \sum_{\ell \geq 1} \mathbf{1}_{\{\bar{J}_\ell = j\}} \delta_{(\bar{T}_\ell, \bar{\Theta}_\ell)}$ is a marked Poisson measure of intensity

$$\bar{\mu}_j(\mathrm{d}t, \mathrm{d}\theta) = \bar{\Lambda} p_j \mathrm{d}t \frac{\mathbf{1}_{\{\theta \leq \bar{\lambda}_j\}}(\theta)}{\bar{\lambda}_j} \mathrm{d}\theta = \mathrm{d}t \mathbf{1}_{\{\theta \leq \bar{\lambda}_j\}}(\theta) \mathrm{d}\theta.$$

As a direct application of Proposition 4.1 , the inhomogeneous multivariate Poisson process is obtained by restriction of each measures $\bar{Q}_j$ to $\Delta_j = \{(t, \theta) \in [0, T] \times [0, \bar{\lambda}_j]; \ \theta \leq \lambda_j(t)\}$ and projection.

**Proposition 4.2.** *The multivariate counting process* $(N^j)_{j \in \mathcal{J}}$, *defined for all* $j \in \mathcal{J}$ *and* $t \in [0, T]$ *by thinning and projection of* $\bar{Q}$:

$$N_t^j := \int_0^t \int_{\mathbb{R}^+} \mathbf{1}_{\{\theta \leq \lambda_j(s)\}} \bar{Q}(\mathrm{d}s, \{j\}, \mathrm{d}\theta) = \sum_{\ell \geq 1} \mathbf{1}_{\{\bar{T}_\ell \leq t\}} \mathbf{1}_{\{\bar{J}_\ell = j\}} \mathbf{1}_{\{\bar{\Theta}_\ell \leq \lambda_j(\bar{T}_\ell)\}},$$

*is an inhomogeneous Poisson process of intensity vector* $(\lambda_j(t))_{j \in \mathcal{J}}$ *on* $[0, T]$.

Proposition 4.2 yields the following simulation Algorithm 1 for multivariate Poisson processes.

---

**Algorithm 1** Thinning algorithm for multivariate inhomogeneous Poisson processes.

1: **Input:** Functions and bounds $(\lambda_j, \bar{\lambda}_j)$, $\lambda_j : [0, T] \to [0, \bar{\lambda}_j]$ and $\bar{\Lambda} = \sum_{j \in \mathcal{J}} \bar{\lambda}_j$
2: **Output:** Points $(T_k, J_k)$ of Poisson measure $N$ on $[0, T] \times \mathcal{J}$
3: Initialization $T_0 \longleftarrow 0$, $\bar{T}_0 \longleftarrow 0$
4: **while** $T_k < T$ **do**
5:      **repeat**
6:          increment iterative variable $\ell \longleftarrow \ell + 1$
7:          compute next proposed time $\bar{T}_\ell \longleftarrow \bar{T}_{\ell-1} + S_\ell$ with $S_\ell \sim \mathscr{E}(\bar{\Lambda})$
8:          draw $\bar{J}_\ell \sim \mathscr{U}\{\bar{\lambda}_j / \bar{\Lambda}, j \in \mathcal{J}\}$ i.e. $\mathbb{P}(\bar{J}_\ell = j) = \bar{\lambda}_j / \bar{\Lambda}$
9:          draw $\bar{\Theta}_\ell \sim \mathscr{U}([0, \bar{\lambda}_{\bar{J}_\ell}])$
10:      **until** accepted event $\bar{\Theta}_\ell \leq \lambda_{\bar{J}_\ell}(\bar{T}_\ell)$
11:      record $(T_k, J_k) \longleftarrow (\bar{T}_\ell, \bar{J}_\ell)$ as accepted point
12: **end while**

---

*Remark* 4.1. The acceptance/rejection Algorithm 1 can be efficient when the functions $\lambda_j$ are of different order, and thus bounded by different $\bar{\lambda}_j$. However, it is important to note that the simulation of the discrete random variables $(\bar{J}_\ell)$ can be costly (compared to a uniform law) when $\mathcal{J}$ is large, for instance when an individual is drawn from a large population. In this case, an alternative is to choose the same bound $\bar{\lambda}_j = \bar{\lambda}$ for all $j \in \mathcal{J}$. Then the marks $(\bar{J}_\ell, \bar{\Theta}_\ell)$ are *i.i.d.* uniform variables on $\mathcal{J} \times [0, \bar{\lambda}]$, faster to simulate.

## 4.2 Simulation algorithm

Let us now come back to the simulation of the IBM introduced in Section 2. For ease of notations, we assume that there are no event with Poisson intensity ($\mathscr{P} = \varnothing$), so that all events that occur are of type

$e \in \mathscr{E} \cup \mathscr{E}_W$, with individual intensity $\lambda_t^e(I, Z_t)$ depending on the population composition $Z_t$ ($e \in \mathscr{E}_W$) or not ($e \in \mathscr{E}$), as defined in Equation 7 and verifying either Assumption 3.2 or Assumption 3.3. The global intensity Equation 9 at time $t \in [0, T]$ is thus

$$\Lambda_t(Z_t) = \sum_{e \in \mathscr{E}} \Big( \sum_{k=1}^{N_t} \lambda^e(t, I_k) \Big) + \sum_{e \in \mathscr{E}_W} \Big( \sum_{k=1}^{N_t} \sum_{j=1}^{N_t} W^e(t, I_k, I_j) \Big) \leq \bar{\Lambda}(N_t), \tag{14}$$

with $\bar{\Lambda}(n) = \big( \sum_{e \in \mathscr{E}} \bar{\lambda}^e \big) n + \big( \sum_{e \in \mathscr{E}_W} \bar{W}^e \big) n^2$.

One of the main difficulty is that the intensity of events is not deterministic as in the case of inhomogeneous Poisson processes, but a function $\Lambda_t(Z_t)$ of the population state, bounded by a function which also depends on the population size. However, the Algorithm 1 can be adapted to simulate the IBM. The construction is done by induction, by conditioning on the state of the population $Z_{T_k}$ at the $k$th event time $T_k$ ($T_0 = 0$).

We first present the construction of the first event at time $T_1$.

## First event simulation

Before the first event time, on $\{t < T_1\}$, the population composition is constant : $Z_t = Z_0 = \{I_1, \dots, I_{N_0}\}$. For each type of event $e$ and individual $I_k$, $k \in \{1, \dots N_0\}$, we denote by $N^{k,e}$ the counting process of intensity $\lambda_t^e(I_k, Z_t)$, counting the occurrences of the events of type $e$ happening to the individual $I_k$. Then, the first event $T_1$ is the first jump time of the multivariate counting vector $(N^{(k,e)})_{(k,e) \in \mathscr{J}_0}$, with $\mathscr{J}_0 = \{1, \dots, N_0\} \times (\mathscr{E} \cup \mathscr{E}_W)$.

Since the population composition is constant before the first event time, each counting process $N^j$ with $j = (k, e) \in \mathscr{J}_0$ coincides on $[0, T_1[$ with an inhomogeneous Poisson process, of intensity $\lambda_t^e(I_k, Z_0)$. Thus (conditionally to $Z_0$), $T_1$ is also the first jump time of an inhomogeneous multivariate Poisson process $N^0 = (N^{0,j})_{j \in \mathscr{J}_0}$ of intensity function $(\lambda_j)_{j \in \mathscr{J}_0}$, defined for all $j = (k, e) \in \mathscr{J}_0$ by:

$$\lambda_j(t) = \lambda_t^e(I_k, Z_0) \leq \bar{\lambda}_0^e \quad \text{with} \quad \bar{\lambda}_0^e = \bar{\lambda}^e \mathbf{1}_{e \in \mathscr{E}} + \bar{W}^e N_0 \mathbf{1}_{e \in \mathscr{E}_W},$$

by Assumption 3.2 and Assumption 3.3. In particular, the jump times of $N^0$ occur at the intensity

$$\Lambda(t) = \sum_{j \in \mathscr{J}_0} \lambda_j(t) = \sum_{e \in \mathscr{E} \cup \mathscr{E}_W} \sum_{k=1}^{N_0} \lambda_t^e(I_k, Z_0) \leq \bar{\Lambda}(N_0) = N_0 \sum_{e \in \mathscr{E} \cup \mathscr{E}_W} \bar{\lambda}_0^e.$$

By Proposition 4.2, $N^0$ can be obtained by thinning of the marked Poisson measure $\bar{Q}^0 = \sum_{\ell \geq 1} \delta_{(\bar{T}_\ell, (\bar{K}_\ell, \bar{E}_\ell), \bar{\Theta}_\ell)}$ on $\mathbb{R}^+ \times \mathscr{J}_0 \times \mathbb{R}^+$, with:

- $(\bar{T}_\ell)_{\ell \in \mathbb{N}^*}$ the jump times of a Poisson process of rate $\bar{\Lambda}(N_0)$.
- $(\bar{K}_\ell, \bar{E}_\ell)_{\ell \in \mathbb{N}^*}$ discrete *i.i.d.* random variables on $\mathscr{J}_0 = \{1, \dots, N_0\} \times (\mathscr{E} \cup \mathscr{E}_W)$, with $K_\ell$ representing the index of the chosen individual and $E_\ell$ the event type for the proposed event, such that:

$$\mathbb{P}(\bar{K}_1 = k, \bar{E}_1 = e) = \frac{\bar{\lambda}_0^e}{\bar{\Lambda}(N_0)} = \frac{1}{N_0} \frac{\bar{\lambda}_0^e N_0}{\bar{\Lambda}(N_0)},$$

i.e. $(\bar{K}_1, \bar{E}_1)$ are distributed as independent random variables where $\bar{K}_1 \sim \mathscr{U}(\{1, \dots, N_0\})$ and $\bar{E}_1$ such that

$$p_e := \mathbb{P}(\bar{E}_1 = e) = \frac{\bar{\lambda}_0^e N_0}{\bar{\Lambda}(N_0)}.$$

- $(\bar{\Theta}_\ell)_{\ell \in \mathbb{N}^*}$ are independent uniform random variables, with $\bar{\Theta}_\ell \sim \mathscr{U}([0, \bar{\lambda}^{\bar{E}_\ell}])$.

Since the first event is the first jump of $N^0$, by Proposition 4.2 and Algorithm 1, the first event time $T_1$ is the first jump time $\bar{T}_\ell$ of $\bar{Q}^0$ such that $\bar{\Theta}_\ell \leq \lambda_{\bar{T}_\ell}^{\bar{E}_\ell}(I_{\bar{K}_\ell}, Z_0)$.

At $T_1 = \bar{T}_\ell$, the event $\bar{E}_\ell$ occurs to the individual $I_{\bar{K}_\ell} = (\tau^b, \infty, x)$. For instance, if $\bar{E}_\ell = d$, a death/exit event occurs, so that $Z_{T_1} = Z_0 + \delta_{(\tau^b, T_1, x)} - \delta_{I_{\bar{K}_\ell}}$ and $N_{T_1} = N_0$. If $\bar{E}_\ell = b$ or $en$, a birth or entry event occurs, so that $N_{T_1} = N_0 + 1$, and a new individual $I_{N_0+1}$ is added to the population, chosen as described in Table 1. Finally, if $\bar{E}_\ell = s$, a swap event occurs, the population size stays constant and $I_{\bar{K}_\ell}$ is replaced by an individual $I'_{\bar{K}_\ell}$, chosen as described in Table 1.

The steps for simulating the first event in the population can be iterated in order to simulate the population. At the $k$th step, the same procedure is repeated to simulate the $k$th event, starting from a population $Z_{T_{k-1}}$ of size $N_{T_{k-1}}$. The algorithm is summarized in Algorithm 2.

---

**Algorithm 2** IBM simulation algorithm (without events of Poissonian intensity)

1: **Input:** Initial population $Z_0$, horizon $T > 0$, and events described by:
2:   - Intensity functions and bounds $(\lambda^e, \bar{\lambda}^e)$ for $e \in \mathscr{E}$ and $(W^e, \bar{W}^e)$ for $e \in \mathscr{E}_W$
3:   - Event action functions $\phi^e(t, I)$ for $e \in \mathscr{E} \cup \mathscr{E}_W$
4: **Output:** Population $Z_T$
5: Initialization $T_0 \longleftarrow 0, \bar{T}_0 \longleftarrow 0$
6: **while** $T_k < T$ **do**
7:     **repeat**
8:         increment iterative variable $\ell \longleftarrow \ell + 1$
9:         compute next proposed time $\bar{T}_\ell \longleftarrow \bar{T}_{\ell-1} + \mathscr{E}(\bar{\Lambda}(N_{T_k}))$
10:        draw a proposed event $\bar{E}_\ell \sim \mathscr{U}\{p_e\}$ with $p_e = \frac{\bar{\lambda}^e \mathbf{1}_{e \in \mathscr{E}} + \bar{W}^e N_{T_k} \mathbf{1}_{e \in \mathscr{E}_W}}{\sum_{e \in \mathscr{E}} \bar{\lambda}^e + \sum_{e \in \mathscr{E}_W} \bar{W}^e N_{T_k}}$
11:        draw an individual index $\bar{K}_\ell \sim \mathscr{U}(\{1, \dots, N_{T_k}\})$
12:        draw $\bar{\Theta}_\ell \sim \mathscr{U}([0, \bar{\lambda}^{\bar{E}_\ell}])$ if $\bar{E}_\ell \in \mathscr{E}$ or $\bar{\Theta}_\ell \sim \mathscr{U}([0, \bar{W}^{\bar{E}_\ell} N_{T_k}])$ if $\bar{E}_\ell \in \mathscr{E}_W$
13:     **until** accepted event $\bar{\Theta}_\ell \leq \lambda_{\bar{T}_\ell}^{\bar{E}_\ell}(I_{\bar{K}_\ell}, Z_{T_k})$
14:     increment iterative variable $k \longleftarrow k + 1$
15:     record $(T_k, E_k, I_k) \longleftarrow (\bar{T}_\ell, \bar{E}_\ell, I_{\bar{K}_\ell})$ as accepted time, event and individual
16:     update the population $Z_{T_k} = Z_{T_{k-1}} + \phi^{E_k}(T_k, I_k)$
17: **end while**

---

**Theorem 4.1.** *A population process $(Z_t)_{t \in [0,T]}$ simulated by the Algorithm 2 is an exact solution of the SDE Equation 12.*

The proof of Theorem 4.1 is detailed in the Appendix Section 8.3.

*Remark* 4.2. The population $Z_{T_k}$ includes dead/exited individuals before the event time $T_k$. Thus, $N_{T_k} > N_{T_k}^a$ is greater than the number of alive individuals at time $T_k$. When a dead individual $I_{\bar{K}_l}$ is drawn from the population during the rejection/acceptance phase of the algorithm, the proposed event $(\bar{T}_\ell, \bar{E}_\ell, I_{\bar{K}_\ell})$ is automatically rejected since the event intensity is $\lambda_{T_\ell}^{\bar{E}_\ell}(I_{\bar{K}_\ell}, Z_{T_k}) = 0$ (nothing can happen to a dead individual). This can slow down the algorithm, especially when the proportion of dead/exited individuals in the population increases. However, the computational cost of keeping dead/exited individuals in the population is much lower than the cost of removing an individual from the population at each death/exit event, which is linear in the population size.

Actually, dead/exited individuals are regularly removed from the population in the `IBMPopSim` algorithm, in order to optimize the trade-off between having to many dead individuals and removing dead individuals from the population too often. The frequency at which dead individuals are "removed

from the population" can be chosen by the user, as an optional argument of the main function `popsim` (see details in Section 4).

*Remark* 4.3. In practice, the bounds $\bar{\lambda}^e$ and $\bar{W}^e$ should be chosen as sharp as possible. It is easy to see that conditionally to $\{\bar{E}_\ell = e, \bar{T}_\ell = t, \bar{K}_\ell = l\}$ the probability of accepting the event is, depending if there are interactions,

$$\mathbb{P}\big(\bar{\Theta}_\ell \leq \lambda_t^e(I_l, Z_{T_k})|\mathscr{F}_{T_k}\big) = \frac{\lambda^e(t, I_l)}{\bar{\lambda}^e}\mathbf{1}_{e \in \mathscr{E}} + \frac{\sum_{j=1}^{N_{T_k}} W^e(t, I_l, I_j)}{\bar{W}^e N_{T_k}}\mathbf{1}_{e \in \mathscr{E}_W}.$$

The sharper the bounds $\bar{\lambda}^e$ and $\bar{W}^e$ are, the higher is the acceptance rate. For even sharper bounds, an alternative is to define bounds $\bar{\lambda}^e(I_l)$ and $\bar{W}^e(I_l)$ depending on the individuals' characteristics. However, the algorithm is modified and the individual $I_l$ is not chosen uniformly in the population anymore. Due to the population size, this is way more costly than choosing uniform bounds, as explained in Remark 4.1.

## 4.3 Simulation algorithm with randomization

Let $e \in E_W$ be an event with interactions. In order to evaluate the individual intensity $\lambda_t^e(I, Z_t) = \sum_{j=1}^{N_t} W^e(t, I, I_j)$ one must compute $W^e(t, I_l, I_j)$ for all individuals in the population. This step can be computationally costly, especially for large populations. One way to avoid this summation is to use randomization (see also Fournier and Méléard (2004) in a model without age). The randomization consists in replacing the summation by an evaluation of the interaction function $W^e$ using an individual $J$ drawn uniformly from the population.

More precisely, if $J \sim \mathscr{U}(\{1, \dots, N_{T_k}\})$ is independent of $\bar{\Theta}_\ell$, we have

$$\mathbb{P}\Big(\bar{\Theta}_\ell \leq \sum_{j=1}^{N_{T_k}} W^e(t, I_l, I_j)|\mathscr{F}_{T_k}\Big) = \mathbb{P}\big(\bar{\Theta}_\ell \leq N_{T_k} W^e(t, I_l, I_J)|\mathscr{F}_{T_k}\big). \tag{15}$$

Equivalently, we can write this probability as $\mathbb{P}\big(\tilde{\Theta}_\ell \leq W^e(t, I_l, I_J)\big)$ where $\tilde{\Theta}_\ell = \frac{\bar{\Theta}_\ell}{N_{T_k}} \sim \mathscr{U}([0, \bar{W}^e])$ is independent of $J \sim \mathscr{U}(\{1, \dots, N_{T_k}\})$.

The efficiency of the randomization procedure increases with the population homogeneity. If the function $W^e$ varies little according to the individuals in the population, the randomization approach is very efficient in practice, especially when the population is large.

We now present the main Algorithm 3 implemented in the `popsim` function of the `IBMPopSim` package age in the case where events arrive with individual intensities, but also with interactions (using randomization) and Poisson intensities. In this general case, $\bar{\Lambda}(n)$ is defined by Equation 11.

---

**Algorithm 3** Randomized IBM simulation algorithm.

1: **Input:** Initial population $Z_0$, horizon $T > 0$, and events described by
2: Intensity functions and bounds $(\lambda^e, \bar{\lambda}^e)$ for $e \in \mathscr{E}$, $(W^e, \bar{W}^e)$ for $e \in \mathscr{E}_W$ and $(\mu^e, \bar{\mu}^e)$ for $e \in \mathscr{P}$
3: Event action functions $\phi^e(t, I)$ for $e \in \mathscr{E} \cup \mathscr{E}_W \cup \mathscr{P}$
4: **Output:** Population $Z_T$
5: Initialization $T_0 \longleftarrow 0$, $\bar{T}_0 \longleftarrow 0$
6: **while** $T_k < T$ **do**
7:     **repeat**
8:         increment iterative variable $\ell \longleftarrow \ell + 1$
9:         compute next proposed time $\bar{T}_\ell \longleftarrow \bar{T}_{\ell-1} + \mathscr{E}\big(\bar{\Lambda}(N_{T_k})\big)$
10:         draw an individual index $\bar{K}_\ell \sim \mathscr{U}(\{1, \dots, N_{T_k}\})$
11:         draw a proposed event $\bar{E}_\ell \sim \mathscr{U}\{p_e\}$ with $p_e = \dfrac{\bar{\mu}^e \mathbf{1}_{e \in \mathscr{P}} + \bar{\lambda}^e N_{T_k} \mathbf{1}_{e \in \mathscr{E}} + \bar{W}^e (N_{T_k})^2 \mathbf{1}_{e \in \mathscr{E}_W}}{\bar{\Lambda}(N_{T_k})}$
12:         **if** $\bar{E}_\ell \in \mathscr{E}$ (without interaction) **then**
13:             draw $\bar{\Theta}_\ell \sim \mathscr{U}\big([0, \bar{\lambda}^{\bar{E}_\ell}]\big)$
14:             *accepted* $\longleftarrow \bar{\Theta}_\ell \leq \lambda^{\bar{E}_\ell}(\bar{T}_\ell, I_{\bar{K}_\ell})$
15:         **end if**
16:         **if** $\bar{E}_\ell \in \mathscr{E}_W$ (with interaction) **then**
17:             draw $(\bar{\Theta}_\ell, J_\ell) \sim \mathscr{U}\big([0, \bar{W}^{\bar{E}_\ell}] \times \{1, \dots, N_{T_k}\}\big)$
18:             *accepted* $\longleftarrow \bar{\Theta}_\ell \leq W^{\bar{E}_\ell}(\bar{T}_\ell, I_{\bar{K}_\ell}, I_{J_\ell})$
19:         **end if**
20:         **if** $\bar{E}_\ell \in \mathscr{P}$ (Poissonian intensity) **then**
21:             draw $\bar{\Theta}_\ell \sim \mathscr{U}\big([0, \bar{\mu}^{\bar{E}_\ell}]\big)$
22:             *accepted* $\longleftarrow \bar{\Theta}_\ell \leq \mu^{\bar{E}_\ell}(\bar{T}_\ell)$
23:         **end if**
24:     **until** *accepted*
25:     increment iterative variable $k \longleftarrow k + 1$
26:     record $(T_k, E_k, I_k) \longleftarrow (\bar{T}_\ell, \bar{E}_\ell, I_{\bar{K}_\ell})$ as accepted time, event and individual
27:     update the population $Z_{T_k} = Z_{T_{k-1}} + \phi^{E_k}(T_k, I_k)$
28: **end while**

---

**Proposition 4.3.** *The population processes $(Z_t)_{t \in [0,T]}$ simulated by the Algorithm 2 and Algorithm 3 have the same law.*

*Proof.* The only difference between Algorithm 2 and Algorithm 3 is in the acceptance/rejection step of proposed events, in the presence of interactions. In Algorithm 3 , a proposed event $(\bar{T}_\ell, \bar{E}_\ell, \bar{K}_\ell)$, with $\bar{E}_l \in \mathscr{E}_W$ (an event with interaction), is accepted as a true event in the population if

$$\bar{\Theta}_\ell \leq W^{\bar{E}_\ell}(\bar{T}_\ell, I_{\bar{K}_\ell}, I_{\bar{J}_\ell}), \text{ with } (\bar{\Theta}_\ell, \bar{J}_\ell) \sim \mathscr{U}\big([0, \bar{W}^{\bar{E}_\ell}] \times \{1, \dots, N_{T_k}\}\big).$$

By Equation 15, the probability of accepting this event is the same than in Algorithm 2 , which achieves the proof. $\qquad\square$

## 5   Model creation and simulation with IBMPopSim

The use of the `IBMPopSim` package is mainly done in two steps: a first model creation followed by the simulation of the population evolution. The creation of a model is itself based on two steps: the description of the population $Z_t$, as introduced in Section 3.1, and the description of the events types, along with their associated intensities, as detailed in Section 3.2 and Section 3.3. A model is compiled

by calling the `mk_model` function, which internally uses a template mechanism to automatically generate the source code describing the model, which is subsequently compiled using the `Rcpp` package to produce the object code.

After the compilation of the model, the simulations are launched by calling the `popsim` function. This function depends on the previously compiled model and simulates a random trajectory of the population evolution based on an initial population and on parameter values, which can change from a call to another.

In this section, we take a closer look at each component of a model in `IBMPopSim`. We also refer to the IBMPopSim website and to the `vignettes` of the package for more details on the package and various examples of model creation.

## 5.1 Population

A population $Z$ is represented by an object of class `population` containing a data frame where each row corresponds to an individual $I = (\tau^b, \tau^d, x)$, and which has at least two columns, `birth` and `death`, corresponding to the birth date $\tau^b$ and death/exit date $\tau^d$ ($\tau^d$ is set to `NA` for alive individuals). The data frame can contain more than two columns if individuals are described by additional characteristics $x = (x_1, \dots x_n)$.

If entry events can occur in the population, the population will contain a characteristic named `entry`. This can be done by setting the flag `entry=TRUE` in the `population` function, or by calling the `add_characteristic` function on an existing population. During the simulation, the date at which an individual enters the population is automatically recorded in the variable `I.entry`. If exit events can occur, the population shall contain a characteristic named `out`. This can be done by setting the flag `out=TRUE` in the `population` function, or by calling the `add_characteristic` function. When an individual `I` exits the population during the simulation, `I.out` is set to `TRUE` and its exit time is recorded as a "death" date.

In the example below, individuals are described by their birth and death dates, as well a Boolean characteristics called male, and the `entry` characteristic. For instance, the first individual is a female whose age at $t_0 = 0$ is 107 and who was originally in the population.

```
pop_init <- population(EW_pop_14$sample, entry=TRUE)
str(pop_init)
```

```
Classes 'population' and 'data.frame':  100000 obs. of  4 variables:
 $ birth: num  -107 -107 -105 -104 -104 ...
 $ death: num  NA NA NA NA NA NA NA NA NA NA ...
 $ male : logi  FALSE FALSE TRUE FALSE FALSE FALSE ...
 $ entry: logi  NA NA NA NA NA NA ...
```

*Individual* In the `C++` model which is automatically generated and compiled, an individual `I` is an object of an internal class containing some attributes (`birth_date`, `death_date` and the characteristics), and some methods including:

- `I.age(t)`: a const method returning the age of an individual `I` at time `t`,
- `I.set_age(a, t)`: a method to set the age `a` at time `t` of an individual `I` (set `birth_date` at `t-a`),
- `I.is_dead(t)`: a const method returning `true` if the individual `I` is dead at time `t`.

*Remark* 5.1. A characteristic $x_i$ must be of atomic type: `logical`, `integer`, `double` or `character`. The function `get_characteristic` allows to easily get characteristics names and their types from a

Table 2: Choices of `CLASS` and `TYPE` arguments for an event creation.

(a) Intensity Classes

| Intensity class | Set | CLASS |
|---|---|---|
| Individual | $\mathscr{E}$ | `individual` |
| Interaction | $\mathscr{E}_W$ | `interaction` |
| Poisson | $\mathscr{P}$ | `poisson` |
| Inhomogeneous Poisson | $\mathscr{P}$ | `inhomogeneous_poisson` |

(b) Event Types

| Event type | TYPE |
|---|---|
| Birth | `birth` |
| Death | `death` |
| Entry | `entry` |
| Exit | `exit` |
| Swap | `swap` |

population data frame. We draw the attention to the fact that some names for characteristics are forbidden, or reserved to specific cases : this is the case for `birth`, `death`, `entry`, `out`, `id`.

## 5.2 Events

The most important step of the model creation is the events creation. The call to the function creating an event is of form

```
mk_event_CLASS(type="TYPE", name="NAME", ...)
```

where `CLASS` is replaced by the class of the event intensity, described in Section 3.3 , and `type` corresponds to the event type, described in Section 3.2. Table 2a and Table 2b summarize the different possible choices for intensity classes and types of event. The optional argument `name` gives a name to the event. If not specified, the name of the event is its type, for instance `death`. However, a name must be specified if the model is composed of several events with the same type (for instance when there are multiple death events corresponding to different causes of death). The other arguments depend on the intensity class and on the event type.

The intensity function and the kernel of an event are defined through arguments of the function `mk_event_CLASS`. These arguments are strings composed of few lines of code. Since the model is compiled using `Rcpp`, the code should be written in `C++`. However, thanks to the functions/variables of the package, even the non-experienced `C++` user can define a model quite easily. To facilitate the implementation, the user can also define a list of **model parameters**, which can be used in the event and intensity definitions. These parameters are stored in a named list and can be of various types: atomic type, numeric vector or matrix, predefined function of one variable (`stepfun`, `linfun`, `gompertz`, `weibull`, `piecewise_x`), piecewise functions of two variables (`piecewise_xy`). We refer to the `vignette(IBMPopSim_cpp)` for more details on parameters types and basic `C++` tools. Another advantage of the model parameters is that their value can be modified from a simulation to another without changing the model.

### 5.2.1 Intensities

In IBMPopSim, the intensity of an event can belong to three classes Section 3.3: individual intensities without interaction between individuals, corresponding to events $e \in \mathscr{E}$, individual intensities with interaction, corresponding to events $e \in \mathscr{E}_W$, and Poisson intensities (homogeneous and inhomogeneous), corresponding to events $e \in \mathscr{P}$.

*Event creation with individual intensity*

An event $e \in \mathscr{E}$ (see Equation 5) has an intensity of the form $\lambda^e(t, I)$ which depends only on the individual `I` and time. Events with such intensity are created using the function

```
mk_event_individual(type = "TYPE",
                    name = "NAME",
                    intensity_code = "INTENSITY", ...)
```

The `intensity_code` argument is a character string containing few lines of C++ code describing the intensity function $\lambda^e(t, I)$. The intensity value has to be stored in a variable called `result` and the available variables for the intensity code are given in Table 3.

Table 3: C++ variables available for intensity code

| Variable | Description |
| --- | --- |
| Variable | Description |
| I | Current individual |
| J | Another individual in the population (only for interaction) |
| t | Current time |
| Model parameters | Depends on the model |

For instance, the intensity code below corresponds to an individual death intensity $\lambda^d(t, I)$ equal to $d_1(a(I,t)) = \alpha_1 \exp(\beta_1 a(I,t))$ for males and $d_2(a(I,t)) = \alpha_2 \exp(\beta_2 a(I,t))$ for females, where $a(I,t) = t - \tau^b$ is the age of the individual $I = (\tau^b, \tau^d, x)$ at time $t$. In this case, the intensity function depends on the individuals' age, gender, and on the model parameters $\alpha = (\alpha_1, \alpha_2)$ and $\beta = (\beta_1, \beta_2)$.

```
death_intensity <- "
    if (I.male) result = alpha_1 * exp(beta_1 * I.age(t));
    else result = alpha_2 * exp(beta_2 * I.age(t));
"
```

*Event creation with interaction intensity*

An event $e \in \mathcal{E}_W$ is an event which occurs to an individual at a frequency which is the result of interactions with other members of the population (see Equation 6), and which can be written as $\lambda_t^e(I, Z_t) = \sum_{J \in Z_t} W^e(t, I, J)$ where $W^e(t, I, J)$ is the intensity of the interaction between individual $I$ and individual $J$.

An event $e \in \mathcal{E}_W$ with such intensity is created by calling the function

```
mk_event_interaction(type = "TYPE",
                     name = "NAME",
                     interaction_code = "INTERACTION_CODE",
                     interaction_type = "random", ...)
```

The `interaction_code` argument contains few lines of C++ code describing the interaction function $W^e(t, I, J)$. The interaction function value has to be stored in a variable called `result` and the available variables for the intensity code are given in Table 3. For example, if we set

```
death_interaction_code <- "result = max(J.size - I.size, 0.);"
```

the death intensity of an individual I is the result of the competition between individuals, depending on a characteristic named `size`, as defined in Equation 8.

The argument `interaction_type`, set by default at `random`, is the algorithm choice for simulating the model. When `interaction_type=full`, the simulation follows Algorithm 2 , `interaction_type=random` it follows Algorithm 3 . In most cases, the `random` algorithm is much faster than the `full` algorithm. For instance in the example of Section 7 the `random` algorithm is 40

times faster on average than the `full` algorithm, on a set of standard parameters. This allows in particular to explore larger parameter sets and population sizes, while avoiding the explosion of computation time.

*Events creation with Poisson and Inhomogeneous Poisson intensity*

For events $e \in \mathscr{P}$ with an intensity $\mu^e(t)$ which does not depend on the population, the event intensity is of class `inhomogeneous_poisson` or `poisson` depending on whether or not the intensity depends on time (in the second case the intensity is constant).

For Poisson (constant) intensities the events are created with the function

```
mk_event_poisson(type = "TYPE",
                 name = "NAME",
                 intensity = "CONSTANT", ...)
```

The following example creates a death event, where individuals die at a constant intensity `lambda` (which has to be in the list of model parameters):

```
mk_event_poisson(type = "death,
                 intensity = "lambda")
```

When the intensity ($\mu^e(t)$) depends on time, the event can be created similarly by using the function

```
mk_event_inhomogeneous_poisson(type = "TYPE",
                               name = "NAME",
                               intensity = "INTENSITY", ...)
```

### 5.2.2 Event kernel code

When an event occurs, the events kernels $k^e$ specify how the event modifies the population. The events kernels are defined in the `kernel_code` parameter of the `mk_event_CLASS(type = "TYPE", name ="NAME", ...)` function. The `kernel_code` is `NULL` by default and doesn't have to be specified for death, exit events and birth events, but mandatory for entry and swap events. Recall that the `kernel_code` argument is a string composed of a few lines of `C++` code, characterizing the individual characteristics following the event. Table 4 summarizes the list of available variables that can be used in the `kernel_code`.

- **Death/Exit event** If the user defines a death event, the death date of the current individual `I` is set automatically to the current time `t`. Similarly, when an individual `I` exits the population, `I.out` is set automatically to `TRUE` and his exit time is recorded as a *death* date. For these events types, the `kernel_code` doesn't have to be specified by the user.

- **Birth event** The default generated event kernel is that an individual `I` gives birth to a new individual `newI` of age `0` at the current time `t`, with same characteristics than the parent `I`. If no kernel is specified, the default generated `C++` code for a birth event is:

```
individual newI = I;
newI.birth_date = t;
pop.add(newI);
```

The user can modify the birth kernel, by specifying the argument `kernel_code` of `mk_event_CLASS`. In this case, the generated code is

```
individual newI = I;
newI.birth_date = t;
```

```
_KERNEL_CODE_
pop.add(newI);
```

where `_KERNEL_CODE_` is replaced by the content of the `kernel_code` argument.

- **Entry event** When an individual `I` enters the population, `I.entry` is set automatically as the date at which the individual enters the population. When an entry occurs the individual entering the population is not of age 0. In this case, the user must specify the `kernel_code` argument indicating how the age and characteristics of the new individual are chosen. For instance, the code below creates an event of type `entry`, named `ev_example`, where individuals enter the population at a Poisson constant intensity. When an individual `newI` enters the population at time `t`, its age is chosen as a normally distributed random variable, with mean 20 and variance 4.

```
mk_event_poisson(
    type = "entry",
    name = "ev_example",
    intensity = "lambda",
    kernel_code = "
        double a_I = max(CNorm(20, 2), 0.);
        newI.set_age(a_I, t);
    ")
```

- **Swap event** The user must specify the `kernel_code` argument indicating how the characteristics of an individual are modified following a swap.

Table 4: `C++` variables available for events kernel code

| Variable | Description |
|---|---|
| Variable | Description |
| `I` | Current individual |
| `J` | Another individual in the population (only for interaction) |
| `t` | Current time |
| `pop` | Current population (vector) |
| `newI` | Available only for birth and entry events. |
| Model parameters | Depends on the model |

When there are several events of the same type, the user can identify which events generated a particular event by adding a characteristic to the population recording the event name/id when it occurs. See e.g. `vignette(IBMPopSim_human_pop)` for an example with different causes of death.

## 5.3 Model creation

Once the population, the events, and model parameters are defined, the IBM model is created using the function `mk_model`.

```
model <- mk_model(characteristics = get_characteristics(pop_init),
                  event = events_list,
                  parameters = model_params)
```

During this step which can take a few seconds, the model is created and compiled using the `Rcpp` package. The model structure in `IBMPopSim` is that the model depends only on the population

characteristics' and parameters names and types, rather than their values. This means that once the model has been created, various simulations can be done with different initial populations and different parameters values.

**Example 5.1.** Here is an example of model with a population structured by age and gender, with birth and death events. The death intensity of an individual of age $a$ is $d(a) = \alpha \exp(\beta a)$, and females between 15 and 40 can give birth with birth intensity $b(a) = \bar{\lambda}^b \mathbf{1}_{[15,40]}$. The newborn is a male with probability $p_{male}$.

```r
params <- list("p_male"= 0.51,
                "birth_rate" = stepfun(c(15,40),c(0,0.05,0)),
                "death_rate" = gompertz(0.008,0.02))

death_event <- mk_event_individual(type = "death", name= "my_death_event",
                intensity_code = "result = death_rate(age(I,t));")

birth_event <- mk_event_individual( type = "birth",
                intensity_code = "if (I.male)
                                        result = 0;
                                    else
                                        result=birth_rate(age(I,t));",
                kernel_code = "newI.male = CUnif(0, 1) < p_male;")
pop <- population(EW_pop_14$sample)

model <- mk_model(characteristics = get_characteristics(pop),
                events = list(death_event,birth_event),
                parameters = params)
```

## 5.4  Simulation

The simulation of the IBM is based on the algorithms presented in Section 4.2 and Section 4.3. The user has first to specify bounds for the intensity or interaction functions of each event type. The random evolution of the population can then be simulated over a period of time $[0, T]$ by calling the function popsim.

*Events bounds*

Since the IBM simulation algorithm is based on an acceptance-rejection method for simulating random times, the user has to specify bounds for the intensity (or interaction) functions of each event (see Assumption 3.2 and Assumption 3.3). These bounds should be stored in a named vector, where for event $e$, the name corresponding to the event bound $\bar{\mu}^e$, $\bar{\lambda}^e$ or $\bar{W}^e$ is the event name defined during the event creation step.

In Example 5.1 from previous section the intensity bound for birth events is $\bar{\lambda}_b$. Since the death intensity function is not bounded, the user will have to specify a maximum age $a_{max}$ in popsim (all individuals above $a_{max}$ die automatically). Then, the bound for death events is $\bar{\lambda}_d = \alpha \exp(\beta a_{max})$. In the example, the death event has been named my_death_event. No name has been specified for the birth event which thus has the default name birth. Then,

```r
a_max <- 120 # maximum age
events_bounds <- c("my_death_event" = params$death_rate(a_max),
                "birth" = max(params$birth_rate))
```

Once the model and events bounds have been defined, a random trajectory of the population can be simulated by calling

```
sim_out <- popsim(model, pop, events_bounds, params,
                  age_max = a_max, time = 30)
```

*Optional parameters*

If there are no events with intensity of class `interaction`, then the simulation can be parallelized easily by setting the optional parameter `multithreading` (`FALSE` by default) to `TRUE`. By default, the number of threads is the number of concurrent threads supported by the available hardware implementation. The number of threads can be set manually with the optional argument `num_threads`. By default, when the proportion of dead individuals in the population exceeds 10%, dead individuals are removed from the current population used in the algorithm (see Remark 4.2). The user can modify this ratio using the optional argument `clean_ratio`, or by removing dead individuals from the population with a certain frequency, given by the `clean_step` argument. Finally, the user can also define the seed of the random number generator stored in the argument `seed`.

*Outputs and treatment of swap events*

The output of the `popsim` function is a list containing three elements: a data frame `population` containing the output population $Z_T$ (or a list of populations $(Z_{t_1}, \dots Z_{t_n})$ if `time` is a vector of times), a numeric vector `logs` of variables related to the simulation algorithm (including the simulation time and number of proposed/accepted events), and the list `arguments` of the simulation inputs, including the initial population, parameters and event bounds used for the simulation.

When there are no swap events (individuals don't change of characteristics), the evolution of the population over the period $[0, T]$ is recorded in a single data frame `sim_out$population` where each line contains the information of an individual who lived in the population over the period $[0, T]$ (see Remark 3.1).

When there are swap events (individuals can change of characteristics), recording the dates of swap events and changes of characteristics following each swap event and for each individual in the population is a memory intensive and computationally costly process. To maintain efficient simulations in the presence of swap events, the argument `time` of `popsim` can be defined as a vector of dates $(t_0, \dots, t_n)$. In this case, `popsim` returns in the object `population` a list of $n$ populations representing the population at time $t_1, \dots t_n$, simulated from the initial time $t_0$. For $i = 1 \dots n$, the $i$th data frame is the population $Z_{t_i}$, i.e. individuals who lived in the population during the period $[t_0, t_i]$, with their characteristics at time $t_i$.

It is also possible to isolate the individuals' life course, by adding an `id` column to the population, which can be done by setting `id=TRUE` in the population construction, or by calling the `add_characteristic` function to an existing population, in order to identify each individual with a unique integer.

Base functions to study the simulation outputs are provided in the package. For instance, the population age pyramid can computed at a given time, as well as death and exposure tables. Several illustrations of the outputs functions are given in the example Section 6 and Section 7.

# 6 Insurance portfolio

This section provides an example of how to use the `IBMPopSim` package to simulate a heterogeneous life insurance portfolio (see also `vignette(IBMPopSim_insurance_portfolio)`).

We consider an insurance portfolio consisting of male policyholders, of age greater than 65. These policyholders are characterized by their age, assumed to be less than $a_{\max} = 110$, and risk class $x \in \mathcal{X} = \{1, 2\}$.

**Entries in the portfolio** New policyholders enter the population at a constant Poisson rate $\mu^{en} = \lambda$, which means that on average, $\lambda$ individuals enter the portfolio each year. A new individual enters the population at an age $a$ that is uniformly distributed between 65 and 70, and is in risk class 1 with probability $p$.

**Death events** A baseline age and time specific death rate is first calibrated on "England and Wales (EW)" males mortality historic data (source: Human Mortality Database https://www.mortality.org/), and projected for 30 years using the Lee-Carter model with the package StMoMo (see A. M. Villegas, Kaishev, and Millossovich (2018)). The forecasted baseline death intensity is denoted by $d(t, a)$, defined by:

$$d(t, a) = \sum_{k=0}^{29} \mathbf{1}_{\{k \le t < k+1\}} d_k(a), \quad \forall\, t \in [0, 30] \text{ and } a \in [65, a_{\max}], \tag{16}$$

with $d_k(a)$ the point estimate of the forecasted mortality rate for age $a$ and year $k$.

Individuals in risk class 1 are assumed to have mortality rates that are 20% higher than the baseline mortality (for instance, the risk class could refer to smokers), while individuals in risk class 2 are assumed to have mortality rates that are 20% lower than the baseline (non smokers). The death intensity of an individual $I = (\tau_b, \infty, x)$, of age $a(I, t) = t - \tau_b$ at time $t$ and in risk class $x \in \{1, 2\}$ is thus the function

$$\lambda^d(t, I) = \alpha_x d(t, a(I, t)), \quad \alpha_1 = 1.2, \quad \alpha_2 = 0.8.$$

In particular, the death intensity verifies Assumption 3.3 since:

$$\lambda^d(t, I) \le \bar{d} := \alpha_1 \sup_{t \in [0, 30]} d(t, a_{\max}). \tag{17}$$

**Exits from the portfolio** Individuals exit the portfolio at a constant (individual) rate $\lambda^{ex}(t, I) = \mu^i$ only depending on their risk class $i \in \{1, 2\}$.

## 6.1 Population

We start with an initial population of 30 000 males of age 65, distributed uniformly in each risk class. The population data frame has thus the two (mandatory) columns `birth` (here the initial time is $t_0 = 0$) and `death` (NA if alive), and an additional column `risk_cls` corresponding to the policyholders risk class. Since there are entry and exit events, the `entry` and `out` flags of the population constructor are set to TRUE.

```
N <- 30000
pop_df <- data.frame("birth" = rep(-65,N), "death" = rep(NA,N),
                     "risk_cls" = rep(1:2,each=N/2))
pop_init <- population(pop_df, entry=TRUE, out=TRUE)
```

## 6.2 Events

**Entry events** The age of the new individual is determined by the `kernel_code` argument in the `mk_event_poisson` function.

```
entry_params <- list("lambda" = 30000, "p" = 0.5)
entry_event <- mk_event_poisson(
    type = "entry",
```

```
        intensity = "lambda",
        kernel_code = "if (CUnif() < p) newI.risk_cls =1;
                      else newI.risk_cls= 2;
                      double a = CUnif(65, 70);
                      newI.set_age(a, t);")
```

Note that the variables `newI` and `t`, as well as the function `CUnif()`, are implicitly defined and usable in the `kernel_code`. The field `risk_cls` comes from the names of characteristics of individuals in the population. The names `lambda` and `p` are parameter names that will be specified in the R named list `params`.

Here we use a constant $\lambda$ as the event intensity, but we could also use a rate $\lambda(t)$ that depends on time, using the function `mk_event_poisson_inhomogeneous`.

**Death and exit events** The baseline death intensity defined in Equation 16 and obtained with the package `StMoMo` is stored in the variable `death_male`.

```
# StMoMo death rates
library('StMoMo')
library('reshape2')
EWStMoMoMale <- StMoMoData(EWdata_hmd, series = "male")
LC <- lc()
ages.fit <- 65:100
years.fit <- 1950:2016
LCfitMale <- fit(LC, data = EWStMoMoMale, ages.fit = ages.fit, years.fit = years.fit)
t <- 30
LCforecastMale <- forecast(LCfitMale, h = t)
d_k <- apply(LCforecastMale$rates, 2, function(x) stepfun(66:100, x))
breaks <- 1:29
death_male <- piecewise_xy(breaks,d_k)
```

The death and exit intensities are of class `individual` (see Table 2a). Hence, the death and exit events are created with the `mk_event_individual` function.

```
death_params <- list("death_male" = death_male, "alpha" = c(1.2, 0.8))
death_event <- mk_event_individual(
    type = "death",
    intensity_code = "result = alpha[I.risk_cls-1] * death_male(t, I.age(t));")

exit_params = list("mu" = c(0.001, 0.06))
exit_event <- mk_event_individual(
    type = "exit",
    intensity_code = "result = mu[I.risk_cls-1]; ")
```

## 6.3   Model creation and simulation

The model is created from all the previously defined building blocks, by calling the `mk_model`.

```
model <- mk_model(
    characteristics = get_characteristics(pop_init),
    events = list(entry_event, death_event, exit_event),
    parameters = c(entry_params, death_params, exit_params))
```

Once the model is compiled, it can be used with different parameters and run simulations for various scenarios. Similarly, the initial population (here `pop_df`) can be modified without rerunning the

mk_model function. The bounds for entry events is simply the intensity $\lambda$. For death events, the bound is given by $\bar{d}$ defined in Equation 17, which is stored in the `death_max` variable.

```
death_max <- max(sapply(d_k, function(x) { max(x) }))
bounds <- c("entry" = entry_params$lambda,
            "death" = death_max,
            "exit" = max(exit_params$mu))

sim_out <- popsim(
    model = model,
    initial_population = pop_init,
    events_bounds = bounds,
    parameters = c(entry_params, death_params, exit_params),
    time = 30,
    age_max = 110,
    multithreading = TRUE)
```

## 6.4  Outputs

The data frame `sim_out$population` consists of all individuals present in the portfolio during the period of $[0, 30]$, including the individuals in the initial population and those who entered the portfolio. Each row represents an individual, with their date of birth, date of death (`NA` if still alive at the end of the simulation), risk class, and characteristics `entry` and `out`. Recall that if an individual enters the population at time $t$, his `entry` characteristic is automatically set up to be equal to $t$. The characteristics `out` is set to `TRUE` for individuals who left the portfolio due to an exit event.

In this example, the simulation time over 30 years, starting from an initial population of 30 000 individuals is very fast (see below), for an acceptance rate of proposed event of approximately 25%. At the end of the simulation, the number of alive individuals is approximately 430 000.

```
[1] "Number of alive individuals in the population at final time T=30 : 426882"
```

```
[1] "Execution time : 0.00017s"
```

```
[1] "Proportion of effective events and proposed events : 0.25"
```

Initially in the portfolio (at $t = 0$), there is the same number of 65 years old policyholders in each risk class. However, policyholders in the risk class 2 with lower mortality rates leave the portfolio at higher rate than policyholders in the risk class 1 : $\mu^2 > \mu^1$. Therefore, the heterogeneous portfolio composition changes with time, including more and more individuals in risk class 1 with higher mortality rates, but with variations across age classes. To illustrate the composition of the total population at the end of the simulation ($t = 30$), we present in Figure 2 the age pyramid of the final composition of the portfolio obtained with the `age_pyramid` and `plot` functions of the `pyramid` class.

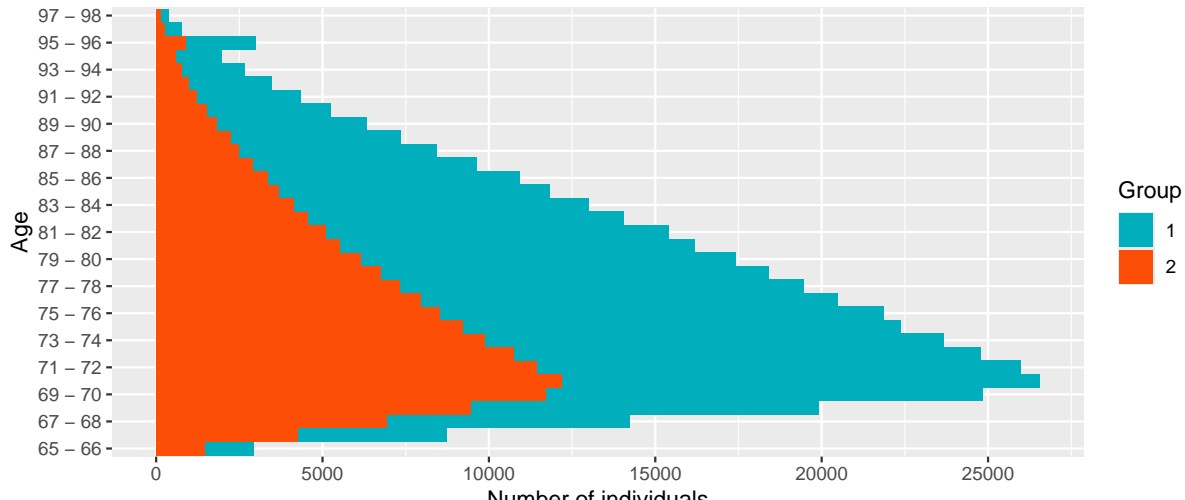

Figure 2: Portfolio age pyramid at t = 30 for individuals in risk class 1 (blue) and 2 (red).

IBMPopSim also allows the fast computation of exact life tables from truncated and censored individual data (due to entry and exit events), using the functions `death_table` and `exposure_table`. These function are particularly efficient, since the computations are made using the `Rccp` library.

```
age_grp <- 65:95
Dx_pop <- death_table(sim_out$population, ages = age_grp, period = 0:30)
Ex_pop <- exposure_table(sim_out$population, ages = age_grp, period = 0:30)
mx_pop <- Dx_pop/Ex_pop
```

In Figure 3, we illustrate the central death rates in the simulated portfolio at final time. Due to the mortality differential between risk class 1 and 2, one would expect to observe more individuals in risk class 2 at higher ages. However, due to exit events, a higher proportion of individuals in risk class 1 exit the portfolio over time, resulting in a greater proportion of individuals in risk class 1 at higher ages than what would be expected in the absence of exit events. Consequently, the mortality rates in the portfolio are more aligned with those of risk class 1 at higher ages. This is a simple example of how composition changes in the portfolio can impact aggregated mortality rates and potentially compensate or reduce an overall mortality reduction (see also (Kaakaï et al. 2019)).

# 7 Population with genetically variable traits

This section provides an example of how to use the IBMPopSim package to simulate an age-structured population with interactions, based on the model proposed in Example 1 of Ferrière and Tran (2009) (see also Méléard and Tran (2009)).

In this model, individuals are characterized by their body size at birth $x_0 \in [0, 4]$ and by their physical age $a \in [0, 2]$. The body size of an individual $I = (\tau^b, \infty, x_0)$ at time $t$ is a linear function of its age $a(I, t) = t - \tau^b$:

$$x(t) = x_0 + ga(I, t),$$

where $g$ is a constant growth rate assumed to be identical for all individuals.

**Birth events** The birth intensity of each individual $I = (\tau^b, \infty, x_0)$ depends on a parameter $\alpha > 0$ and on its initial size, as given by the equation

$$\lambda^b(t, I) = \alpha(4 - x_0) \leq \bar{\lambda}^b = 4\alpha. \tag{18}$$

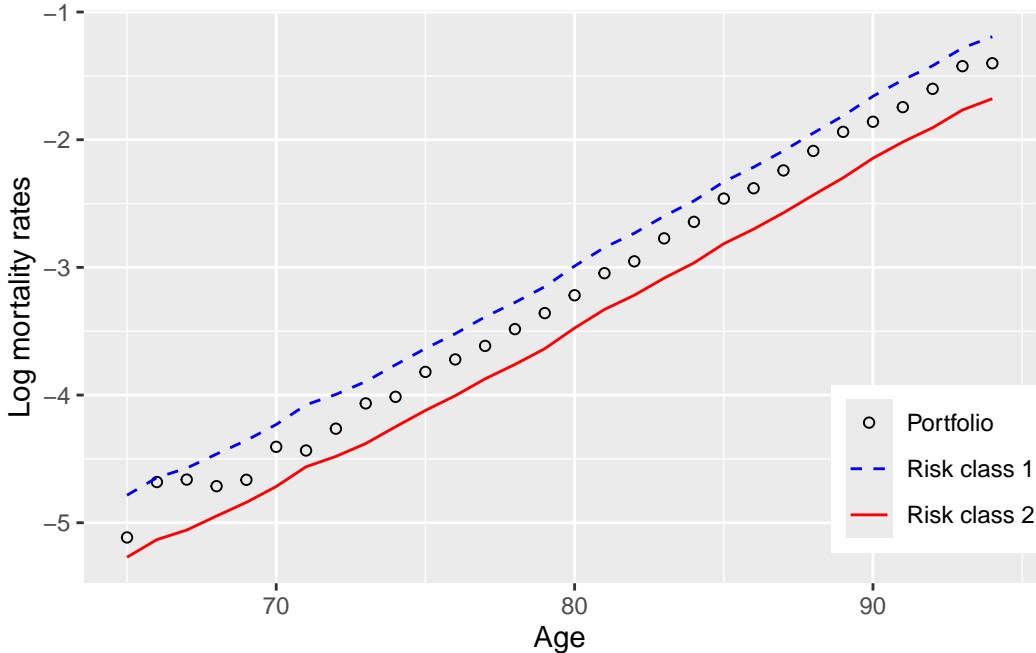

Figure 3: Portfolio central death rates at t = 30 (black).

Thus, smaller individuals have a higher birth intensity. When a birth occurs, the new individual inherits the same birth size $x_0$ as its parent with high probability $1 - p$, or a mutation can occur with probability $p$, resulting in a birth size given by

$$x_0' = \min(\max(0, x_0 + G), 4), \tag{19}$$

where $G$ is a Gaussian random variable with mean 0 and variance $\sigma^2$.

**Death events** Due to competition between individuals, the death intensity of an individual depends on the size of other individuals in the population. Bigger individuals have a better chance of survival. If an individual $I = (\tau^b, \infty, x_0)$ of size $x(t) = x_0 + ga(I, t)$ encounters an individual $J = (\tau_J^b, \infty, x_0')$ of size $x'(t) = x_0' + ga(J, t)$, then it can die with the intensity

$$W(t, I, J) = U(x(t), x'(t)),$$

where the interaction function $U$ is defined by

$$U(x, y) = \beta\left(1 - \frac{1}{1 + c\exp(-4(x - y))}\right) \leq \bar{W} = \beta. \tag{20}$$

The death intensity of an individual $I$ at time $t$ and in a population $Z$ is the result of interactions with all individuals in the population, including itself, and is given by

$$\lambda_t^d(I, Z) = \sum_{J=(\tau^b, \infty, x_0')\in Z} W(x_0 + ga(I, t), x_0' + ga(J, t)),$$

## 7.1 Population

We use an initial population of 900 living individuals, all of whom have the same size and ages uniformly distributed between 0 and 2 years.

```
N <- 900
x0 <- 1.06
agemin <- 0.
agemax <- 2.

pop_df <- data.frame(
  "birth" = -runif(N, agemin, agemax), # Uniform age in [0,2]
  "death" = as.double(NA), # All individuals are alive
  "birth_size" = x0) # All individuals have the same initial birth size x0
pop_init <- population(pop_df)
```

## 7.2 Events

### 7.2.1 Birth events

The parameters involved in a birth event are the probability of mutation $p$, the variance of the Gaussian random variable and the coefficient $\alpha$ of the intensity.

```
params_birth <- list("p" = 0.03, "sigma" = sqrt(0.01), "alpha" = 1)
```

The birth intensity Equation 18 is of class `individual`. Hence, the event is created by calling the `mk_event_individual` function. The size of the new individual is given in the kernel following Equation 19.

```
birth_event <- mk_event_individual(
  type = "birth",
  intensity_code = "result = alpha*(4 - I.birth_size);",
  kernel_code = "if (CUnif() < p)
                    newI.birth_size = min(max(0.,CNorm(I.birth_size,sigma)),4.);
                  else
                    newI.birth_size = I.birth_size;")
```

### 7.2.2 Death events

The death intensity Equation 20 is of class `interaction`. Hence, the event is created by calling the `mk_event_interaction` function. The parameters used for this event are the growth rate $g$, the amplitude of the interaction function $\beta$, and the strength of competition $c$.

```
params_death <- list("g" = 1, "beta" = 2./300., "c" = 1.2)
death_event <- mk_event_interaction(
  type = "death",
  interaction_code = "double x_I = I.birth_size + g * age(I,t);
                      double x_J = J.birth_size + g * age(J,t);
                      result = beta*(1.-1./(1.+c*exp(-4.*(x_I-x_J))));")
```

## 7.3 Model creation and simulation

The model is created using the `mk_model` function.

```
model <- mk_model(
    characteristics = get_characteristics(pop_init),
    events = list(birth_event, death_event),
    parameters = c(params_birth, params_death))
```

The simulation of one scenario can then be launched with the call of the `popsim` function, after computing the events bounds $\bar{\lambda}^b = 4\alpha$ and $\bar{W} = \beta$.

```
sim_out <- popsim(model = model,
    initial_population = pop_init,
    events_bounds = c("birth" = 4 * params_birth$alpha,
                       "death" = params_death$beta),
    parameters = c(params_birth, params_death),
    age_max = 2,
    time = 500)
```

Based on the results of a simulation, we can reproduce the numerical results of Ferrière and Tran (2009). In Figure 4, we draw a line for each individual in the population to represent their birth size during their lifetime.

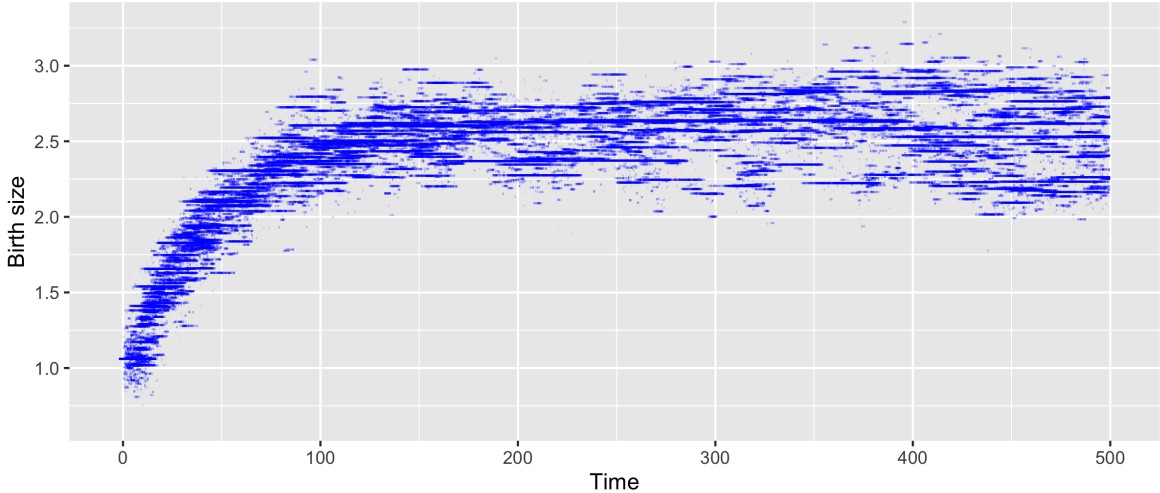

Figure 4: Evolution of birth size

In this example, the randomized Algorithm 3 allows for much faster computation times than the model implemented below with Algorithm 2 ("full" algorithm):

```
death_event_full <- mk_event_interaction(type = "death",
    interaction_type= "full",
    interaction_code = "double x_I = I.birth_size + g * age(I,t);
                        double x_J = J.birth_size + g * age(J,t);
                        result = beta * ( 1.- 1./(1. + c * exp(-4. * (x_I-x_J))));"
)

model_full <- mk_model(characteristics = get_characteristics(pop_init),
    events = list(birth_event, death_event_full),
    parameters = c(params_birth, params_death))

sim_out_full <- popsim(model = model_full,
    initial_population = pop_init,
    events_bounds =c("birth" = 4 * params_birth$alpha, "death" = params_death$beta),
    parameters = c(params_birth, params_death),
    age_max = 2,
    time = 500)
```

```
[1] "The full algorithm is 36 times slower than the randomized version"
```

In Figure 5, the two algorithms are compared for different population sizes. We progressively decrease the value of the mortality rate parameter $\beta$ and increase the birth rate parameter $\alpha$. Starting with the values provided in Ferrière and Tran (2009), $\alpha = 1$ and $\beta = 2/300$, resulting in a stationary population size of approximately $N = 360$ individuals for a sample of 50 simulations, we can easily increase the stationary population size to approximately $N = 2600$ individuals with $\alpha = 2$ and $\beta = 1/300$.[4] In the log-scaled figure, we can observe the trend of computation time as a function of the population size $N$, which is linear for the randomized algorithm and quadratic for the full one ( Algorithm 2 ). We can also see that the randomized version of the algorithm is between 17 to 100 times faster than the full one in this example, taking only 2 seconds in average for the randomized version versus 211 seconds for Algorithm 2 for the biggest population size ($N = 2600$) and $T = 500$.

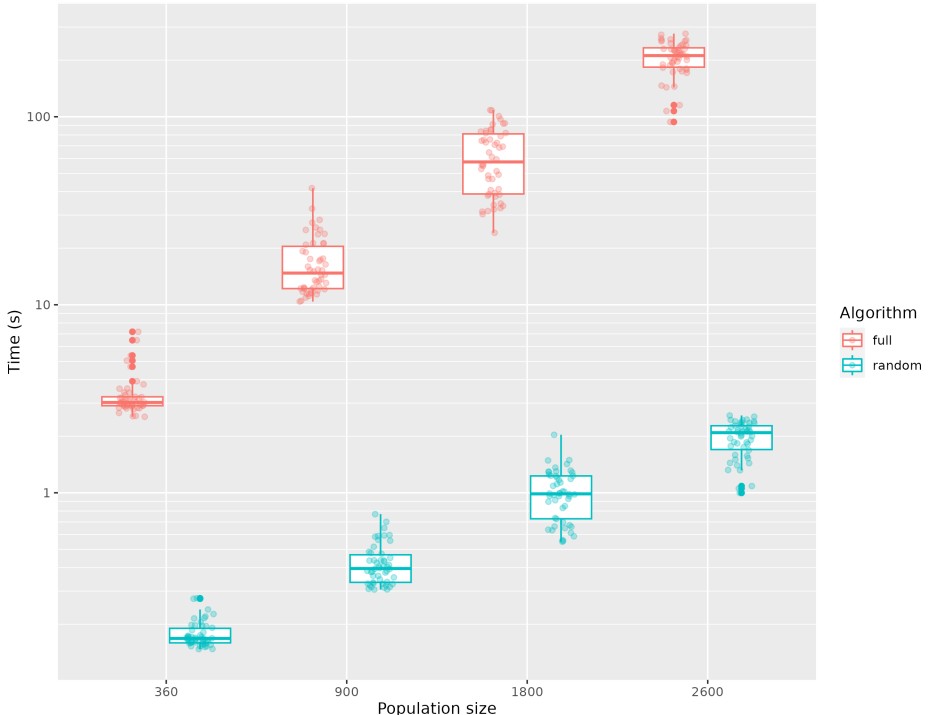

Figure 5: Full vs random algorithm computation time

# 8   Appendix

## 8.1   Recall on Poisson random measures

We recall below some useful properties of Poisson random measures, mainly following Chapter 6 of (Çinlar 2011). We also refer to (Kallenberg 2017) for a more comprehensive presentation of random counting measures.

**Definition 8.1** (Poisson Random Measures). Let $\mu$ be a $\sigma$-finite diffuse measure on a Borel subspace $(E, \mathscr{E})$ of $(\mathbb{R}^d, \mathscr{B}(\mathbb{R}^d))$. A random counting measure $Q = \sum_{k \geq 1} \delta_{X_k}$ is a Poisson (counting) random measure of *mean measure* $\mu$ if

---

[4]The choices $(\alpha, \beta) \in \{(1, 2/300), (1, 1/300), (1.5, 1/300), (2, 1/300)\}$ lead to the stationary population sizes $N \in \{360, 900, 1800, 2600\}$. For each set of parameters, we generated a new initial population, which was used for a benchmark of 50 simulations with both randomized and full algorithm. The simulations run on a Intel Core i7-8550U CPU 1.80GHz × 8 processor, with 15.3 GiB of RAM, under Debian GNU/Linux 11.

1. $\forall A \in \mathscr{E}$, $Q(A)$ is a Poisson random variable with $\mathbb{E}[Q(A)] = \mu(A)$.
2. For all disjoints subsets $A_1, \ldots, A_n \in \mathscr{E}$, $Q(A_1), \ldots, Q(A_n)$ are independent Poisson random variables.

Let us briefly recall here some simple but useful operations on Poisson measures. In the following, $Q$ is a Poisson measure of mean measure $\mu$, unless stated otherwise.

**Proposition 8.1** (Restricted Poisson measure). *If $B \in \mathscr{E}$, then, the restriction of $Q$ to $B$ defined by*

$$Q^B = \mathbf{1}_B Q = \sum_{k \geq 1} \mathbf{1}_B(X_k) \delta_{X_k}$$

*is also a Poisson random measure, of mean measure $\mu^B = \mu(\cdot \cap B)$.*

**Proposition 8.2** (Projection of Poisson measure). *If $E = F_1 \times F_2$ is a product space, then the projection*

$$Q_1(\mathrm{d}x) = \int_{F_2} Q(\mathrm{d}x, \mathrm{d}y)$$

*is a Poisson random measure of mean measure $\mu_1(\mathrm{d}x) = \int_{F_2} \mu(\mathrm{d}x, \mathrm{d}y)$.*

### 8.1.1 Link with Poisson processes

Let $Q = \sum_{k \geq 1} \delta_{T_k}$ a Poisson random measure on $E = \mathbb{R}^+$ with mean measure $\mu(\mathrm{d}t) = \Lambda(t)\mathrm{d}t$ absolutely continuous with respect to the Lebesgue measure, $\mu(A) = \int_A \Lambda(t)\mathrm{d}t$. The counting process $(N_t)_{t \geq 0}$ defined by

$$N_t = Q([0, t]) = \sum_{k \geq 1} \mathbf{1}_{\{T_k \leq t\}}, \quad \forall t \geq 0, \tag{21}$$

is an inhomogeneous Poisson process with intensity function (or rate) $t \mapsto \Lambda(t)$. In particular, when $\Lambda(t) \equiv c$ is a constant, $N$ is a homogeneous Poisson process with rate $c$. Assuming that the atoms are ordered $T_1 < T_2 < \ldots$, we recall that the sequence $(T_{k+1} - T_k)_{k \geq 1}$ is a sequence of *i.i.d.* exponential variables of parameter $c$.

### 8.1.2 Marked Poisson measures on $E = \mathbb{R}^+ \times F$

We are interested in the particular case when $E$ is the product space $\mathbb{R}^+ \times F$, with $(F, \mathscr{F})$ a Borel subspace of $\mathbb{R}^d$. Then, a random counting measure is defined by a random set $S = \{(T_k, \Theta_k), k \geq 1\}$. The random variables $T_k \geq 0$ can be considered as time variables, and constitute the jump times of the random measure, while the variables $\Theta_k \in F$ represent space variables.

We recall in this special case the Theorem VI.3.2 in (Çinlar 2011).

**Proposition 8.3** (Marked Poisson measure). *Let $m$ be a $\sigma$−finite diffuse measure on $\mathbb{R}^+$, and $K$ a transition probability kernel from $(\mathbb{R}^+, \mathscr{B}(\mathbb{R}^+))$ into $(F, \mathscr{F})$. Assume that the collection $(T_k)_{k \geq 1}$ forms a Poisson process $(N_t) = (\sum_{k \geq 1} \mathbf{1}_{\{T_k \leq t\}})$ with mean $m(\mathrm{d}t) = \Lambda(t)\mathrm{d}t$, and that given $(T_k)_{k \geq 1}$, the variables $\Theta_k$ are conditionally independent and have the respective distributions $K(T_k, \cdot)$.*

1. *Then, $\{(T_k, \Theta_k); k \geq 1\}$ forms a Poisson random measure $Q = \sum_{k \geq 1} \delta_{(T_k, \Theta_k)}$ on $(\mathbb{R}^+ \times F, \mathscr{B}(\mathbb{R}^+) \otimes \mathscr{F})$, called a* Marked point process *, with mean $\mu$ defined by*

$$\mu(\mathrm{d}t, \mathrm{d}y) = \Lambda(t)\mathrm{d}t K(t, \mathrm{d}y).$$

2. *Reciprocally let $Q$ be a Poisson random measure of mean measure $\mu(\mathrm{d}t, \mathrm{d}y)$, admitting the following disintegration with respect to the first coordinate: $\mu(\mathrm{d}t, \mathrm{d}y) = \tilde{\Lambda}(t)\mathrm{d}t \nu(t, \mathrm{d}y)$, with $\nu(t, F) < \infty$. Let $K(t, \mathrm{d}y) = \dfrac{\nu(t, \mathrm{d}y)}{\nu(t, F)}$ and $\Lambda(t) = \nu(t, F)\tilde{\Lambda}(t)$. Then, $Q = \sum_{k \geq 1} \delta_{(T_k, \Theta_k)}$ is a marked Poisson*

measure with $(T_k, \Theta_k)_{k \in \mathbb{N}^*}$ defined as above. In particular, the projection $N = (N_t)_{t \geq 0}$ of the Poisson measure on the first coordinate,

$$N_t = Q([0,t] \times F) = \sum_{k \geq 1} \mathbf{1}_{[0,t] \times F}(T_k, \Theta_k) = \sum_{k \geq 1} \mathbf{1}_{\{T_k \leq t\}}, \quad \forall \, t \geq 0,$$

is an inhomogeneous Poisson process of rate $\Lambda(t) = \nu(t, F)\tilde{\Lambda}(t)$.

When the transition probability kernel $K$ does not depend on the time: $K(t, A) = \nu(A)$ for some probability measure $\nu$, then the marks $(\Theta_k)_{k \geq 1}$ form an *i.i.d.* sequence with distribution $\nu$, independent of $(T_k)_{k \geq 1}$.

The preceding proposition thus yields a straightforward iterative simulation procedure for a Marked Poisson process on $[0, T] \times F$ with mean measure $\mu(\mathrm{d}t, \mathrm{d}y) = c\,\mathrm{d}t\,K(t, \mathrm{d}y)$ and $c > 0$. The procedure is described in Algorithm 4 .

---

**Algorithm 4** Simulation of Marked Poisson measure

---

1: **Input:** Constant $c$, simulatable kernel $K$ and final time $T$
2: **Output:** Times $(T_1, \dots, T_n)$ and Marks $(Y_1, \dots, Y_n)$ of the Marked Poisson measure of mean $\mu(\mathrm{d}t, \mathrm{d}y) = c\,\mathrm{d}t\,K(t, \mathrm{d}y)$ in $[0, T] \times F$.
3: Initialization draw $T_1 \sim \mathscr{E}(c)$ and draw $Y_1 \sim K(T_1, \mathrm{d}y)$
4: **while** condition **do**
5:      increment iterative variable $k \longleftarrow k + 1$
6:      compute next jump time compute next jump time $T_k \longleftarrow T_{k-1} + \mathscr{E}(c)$
7:      draw a conditional mark $Y_k \sim K(T_k, \mathrm{d}y)$
8: **end while**

---

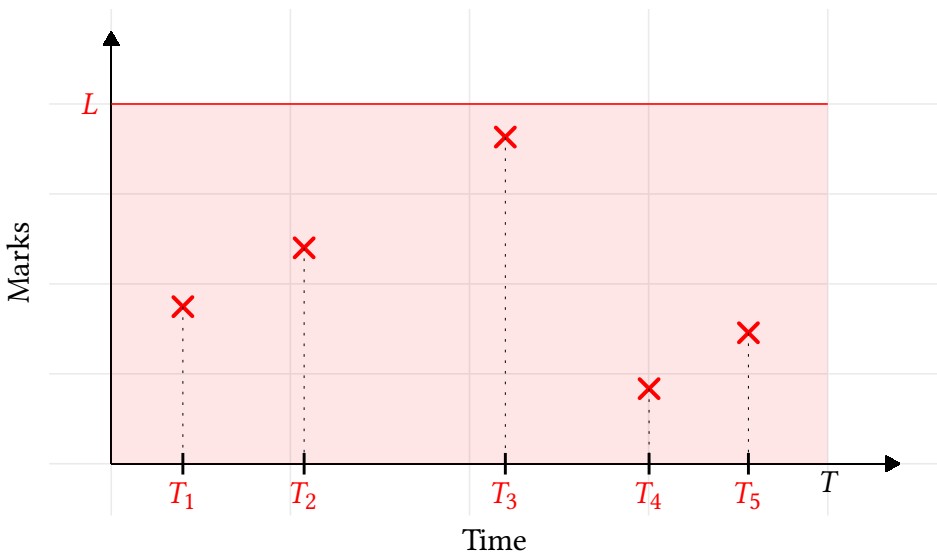

Figure 6: Example of Marked Poisson measure on $[0, T]$ with $m(\mathrm{d}t) = L\,\mathrm{d}t$ (jump times occur at Poisson arrival times of rate $L$) and with $\nu(\mathrm{d}y) = \frac{1}{L}\mathbf{1}_{[0,L]}(y)\mathrm{d}y$ (marks are drawn uniformly on $[0, L]$). The mean measure is then $\mu(\mathrm{d}t, \mathrm{d}y) = \mathrm{d}t\mathbf{1}_{[0,L]}(y)\mathrm{d}y$.

## 8.2   Pathwise representation of IBMs

**Notation reminder** The population's evolution is described by the measure valued process $(Z_t)_{t \geq 0}$. Several types of events $e$ can occur to individuals denoted by $I$. If an event of type $e$ occur to the

individual $I$ at time $t$, then the population state $Z_{t^-}$ is modified by $\phi^e(t, I)$. If $e \in \mathcal{E} \cup \mathcal{E}_W$, then events of type $e$ occur with an intensity $\sum_{k=1}^{N_t} \lambda_t^e(I, Z_t)$, with $\lambda_t^e(I, Z_t)$ defined by Equation 7. If $e \in \mathscr{P}$, then events of type $e$ occur in the population at a Poisson intensity of $(\mu_t^e)$.

### 8.2.1 Proof of Theorem 3.1

*Proof.* For ease of notation, we prove the case when $\mathscr{P} = \varnothing$ (there are no events with Poisson intensity).

- Step 1. The existence of a solution to Equation 12 is obtained by induction. Let $Z^1$ be the unique solution the thinning equation:

$$Z_t^1 = Z_0 + \int_0^t \int_{\mathcal{J} \times \mathbb{R}^+} \phi^e(s, I_k) \mathbf{1}_{\{k \leq N_0\}} \mathbf{1}_{\{\theta \leq \lambda_s^e(I_k, Z_0)\}} Q(\mathrm{d}s, \mathrm{d}k, \mathrm{d}e, \mathrm{d}\theta), \quad \forall 0 \leq t \leq T.$$

  Let $T_1$ be the first jump time of $Z^1$. Since $Z_s^1 = Z_0$ and $N_{s^-} = N_0$ on $[0, T_1]$, $Z^1$ is solution of Equation 12 on $[0, T_1]$.

Let us now assume that Equation 12 admits a solution $Z^n$ on $[0, T_n]$, with $T_n$ the $n$–th event time in the population. Let $Z^{n+1}$ be the unique solution of the thinning equation:

$$Z_t^{n+1} = Z_{t \wedge T_n}^n + \int_{t \wedge T_n}^t \int_{\mathcal{J} \times \mathbb{R}^+} \phi^e(s, I_k) \mathbf{1}_{\{\theta \leq \lambda_s^e(I_k, Z_{T_n}^n)\}} \mathbf{1}_{\{k \leq N_{T_n}^n\}} Q(\mathrm{d}s, \mathrm{d}k, \mathrm{d}e, \mathrm{d}\theta).$$

First, observe that $Z^{n+1}$ coincides with $Z^n$ on $[0, T_n]$. Let $T_{n+1}$ be the $(n + 1)$–th jump of $Z^{n+1}$. Furthermore, $Z_s^{n+1} = Z_{T_n}^n$ and $N_s^{n+1} = N_{T_n}^n$ on $[T_n, T_{n+1}]$ (nothing happens between two successive event times), $Z^{n+1}$ verifies for all $t \leq T_{n+1}$:

$$Z_t^{n+1} = Z_{t \wedge T_n}^n + \int_{t \wedge T_n}^t \int_{\mathcal{J} \times \mathbb{R}^+} \phi^e(s, I_k) \mathbf{1}_{\{\theta \leq \lambda_s^e(I_k, Z_{s^-}^{n+1})\}} \mathbf{1}_{\{k \leq N_s^{n+1}\}} Q(\mathrm{d}s, \mathrm{d}k, \mathrm{d}e, \mathrm{d}\theta).$$

Since, $Z^n$ is a solution of Equation 12 on $[0, T_n]$ coinciding with $Z^{n+1}$ this achieves to prove that $Z^{n+1}$ is solution of Equation 12 on $[0, T_{n+1}]$. Finally, let $Z = \lim_{n \to \infty} Z^n$. For all $n \geq 1$, $T_n$ is the $n$–th event time of $Z$, and $Z$ is solution of Equation 12 on all time intervals $[0, T_n \wedge T]$ by construction.

By Lemma 3.1 $T_n \xrightarrow[n \to \infty]{} \infty$. Thus, by letting $n \to \infty$ we can conclude that $Z$ is a solution of Equation 12 on $[0, T]$.

- Step 2. Let $\tilde{Z}$ be a solution of Equation 12. Using the same arguments than in Step 1, it is straightforward to show that $\tilde{Z}$ coincides with $Z^n$ on $[0, T_n]$, for all $n \geq 1$. Thus, $\tilde{Z} = Z$, with achieves to prove uniqueness.

$\square$

### 8.2.2 Proof of Lemma 3.1

The proof is obtained using pathwise comparison result, generalizing those obtained in (Kaakai and El Karoui 2023).

*Proof.* Let $Z$ be a solution of Equation 12. For all $e \in \mathscr{P} \cup \mathcal{E} \cup \mathcal{E}_W$, let $N^e$ be the process counting the occurrence of events of type $e$ in the population. $N^e$ is a counting process of $\{\mathscr{F}_t\}$-intensity $(\Lambda_t^e(Z_{t^-}))$, solution of

$$
\begin{aligned}
N_t^e &= \int_0^t \int_{\mathbb{N} \times \mathbb{R}^+} \mathbf{1}_{\{k \leq N_{s^-}\}} \mathbf{1}_{\{\theta \leq \lambda_s^e(I_k, Z_{s^-})\}} Q(\mathrm{d}s, \mathrm{d}k, \{e\}, \mathrm{d}\theta), && if\, e \in \mathcal{E} \cup \mathcal{E}_W, \\
N_t^e &= \int_0^t \int_{\mathbb{R}^+} \mathbf{1}_{\{\theta \leq \mu_s^e\}} Q^{\mathscr{P}}(\mathrm{d}s, \{e\}, \mathrm{d}\theta), && if\, e \in P.
\end{aligned}
\tag{22}
$$

By definition, the jump times of the multivariate counting process $(N^e)_{e \in \mathcal{P} \cup \mathcal{E} \cup \mathcal{E}_W}$ are the population event times $(T_n)_{n \geq 0}$. The idea of the proof is to show that $(N^e)_{e \in \mathcal{P} \cup \mathcal{E} \cup \mathcal{E}_W}$ does not explode in finite time, by pathwise domination with a simpler multivariate counting process. The first steps are to control the population size $N_t = N_0 + N_t^b + N_t^{en}$.

**Step 1** Let $(\bar{N}^b, \bar{N}^{en})$ be the 2-dimensional counting process defined as follows: for $e \in \{b, en\}$, $\bar{N}_0^e = 0$ and

$$
\begin{aligned}
\bar{N}_t^e &= \int_0^t \int_{\mathbb{N} \times \mathbb{R}^+} \mathbf{1}_{\{k \leq N_0 + \bar{N}_{s-}\}} \mathbf{1}_{\{\theta \leq f^e(N_0 + \bar{N}_{s-})\}} Q(\mathrm{d}s, \mathrm{d}k, \{e\}, \mathrm{d}\theta), \quad & if\, e \in \mathcal{E} \cup \mathcal{E}_W, \\
\bar{N}_t^e &= \int_0^t \int_{\mathbb{R}^+} \mathbf{1}_{\{\theta \leq \bar{\mu}^e\}} Q^{\mathcal{P}}(\mathrm{d}s, \{e\}, \mathrm{d}\theta) \quad & if\, e \in P,
\end{aligned}
\tag{23}
$$

with $\bar{N} := \bar{N}^b + \bar{N}^{en}$ and $f^e$ the function introduced in Assumption 3.4.

- If $b, en \in P$, then $\bar{N}$ is a inhomogeneous Poisson process.

- If $b, en \in \mathcal{E} \cup \mathcal{E}_W$, then it is straightforward to show that conditionally to $N_0$, $\bar{N}$ is a pure birth Markov process with birth intensity function $g(n) = n\big(f^b(N_0 + n) + f^{en}(N_0 + n)\big)$. In particular, by Assumption 3.4, $g$ verifies the standard Feller condition for pure birth Markov processes (see e.g. (Bansaye and Méléard 2015)):

$$
\sum_{n=1}^{\infty} \frac{1}{g(n)}.
$$

- Finally, if $b \in \mathcal{E}$ and $en \in P$ (or equivalently if $b \in P$ and $en \in \mathcal{E}$), then one can show easily that $\bar{N}$ is a pure birth Markov process with immigration, of birth intensity function $g(n) = \bar{\mu}^{en} + nf^b(N_0 + n)$ (resp. $g(n) = \bar{\mu}^b + nf^{en}(N_0 + n)$), also verifying the Feller condition. Therefore, there exists a non-exploding solution of Equation 23, by Proposition 3.3 in (Kaakai and El Karoui 2023).

**Step 2** The second step consists in showing that $(N^b, N^{en})$ is strongly dominated by $(\bar{N}^b, \bar{N}^{en})$, i.e that all jumps of $(N^b, N^{en})$ are jumps of $(\bar{N}^b, \bar{N}^{en})$. Without loss of generality, we can assume that $f^e : \mathbb{N} \to (0, +\infty)$ is increasing since $f^e(n)$ can be replaced by $\sup_{\{m \leq n\}} f^e(m)$.

Let $e \in \{b, en\}$. If $e \in \mathcal{P}$, then for all $s \in [0, T]$

$$
\{\theta \leq \mu_s^e\} \subset \{\theta \leq \bar{\mu}^e\},
$$

which yields that all jumps of $N^e$ are jumps of $\bar{N}^e$.

If $e \in \mathcal{E} \cup \mathcal{E}_W$, the proof by induction is analogous to the proof of Proposition 2.1 in (Kaakai and El Karoui 2023). Let $T_1^e$ be first jump time of $N^e$, associated with the marks $(K_1^e, \Theta_1^e)$ of $Q$ (or $Q^{\mathcal{P}}$). Then, by Definition of Equation 22, $K_1^e \leq N_0$ and $\Theta_1^e \leq \lambda_{T_1^e}^e(I_{K_1^e}, Z_0)$.

By Assumption 3.4, we have also

$$
\Theta_1^e \leq \lambda_{T_1^e}^e(I_{K_1^e}, Z_0) \leq f^e(N_0) \leq f^e(N_0 + \bar{N}_{T_1^{e,-}}), \quad K_1^e \leq N_0 + \bar{N}_{T_1^{e,-}}.
$$

Thus, $T_1^e$ is also a jump time of $\bar{N}^e$. By iterating this argument, we obtain that all jump times of $N^e$ are jump times of $\bar{N}^e$.

Thus, $(N^b, N^{en})$ does not explode in finite time.

**Step 3** It remains to show that for $e \notin \{b, en\}$, $N^e$ does not explode.

Let $e \neq b, en$. If $e \in \mathcal{P}$, the proof is the same than in Step 2. Otherwise, let:

$$
h_t^e(n) = \sup_{I \in \mathcal{I}, m \leq n} \lambda_t^e\Big(I, \sum_{k=1}^m \delta_{I_k}\Big), \quad \forall\, t \in [0, T]\, n \in \mathbb{N}^*.
$$

By Assumption 3.2 and Assumption 3.3, $h_t^e(n) < \infty$, and we can introduce the non exploding counting process $\bar{N}^e$, defined by the thinning equation :

$$\bar{N}_t^e = \int_0^t \int_{\mathbb{N} \times \mathbb{R}^+} \mathbf{1}_{\{k \leq N_0 + \bar{N}_{s^-}\}} \mathbf{1}_{\{\theta \leq h_s^e(N_0 + \bar{N}_{s^-})\}} Q(\mathrm{d}s, \mathrm{d}k, \{e\}, \mathrm{d}\theta),$$

with $\bar{N}_s = \bar{N}_s^b + \bar{N}_s^{en}$.

Finally, by Step 2, for $s \in [0, T]$ the population size $N_s = N_0 + N_s^b + N_s^{en}$ is bounded a.s. by $N_0 + \bar{N}_s$, since all jumps of $(N^b, N^{en})$ are jumps of $(\bar{N}^b, \bar{N}^{en})$. Thus, for all $s \in [0, T]$,

$$\{k \leq N_{s^-}\} \subset \{k \leq N_0 + \bar{N}_{s^-}\}, \text{ and } \{\theta \leq \lambda_s^e(I_k, Z_{s^-})\} \subset \{\theta \leq h_s^e(N_0 + \bar{N}_{s^-})\}.$$

This proves that all jumps of $N^e$ are jumps $\bar{N}^e$, and thus $N^e$ does not explode in finite time. □

### 8.2.3 Alternative pathwise representation

**Theorem 8.1.** *Let $\mathscr{J}_{\mathscr{E}} = \mathbb{N} \times \mathscr{E}$ and $\mathscr{J}_W = \mathbb{N} \times \mathscr{E}_W$.*

*Let $Q^{\mathscr{E}}$ be a random Poisson measure on $\mathbb{R}^+ \times \mathscr{J}_{\mathscr{E}} \times \mathbb{R}^+$, of intensity $\mathrm{d}t \delta_{\mathscr{J}_{\mathscr{E}}}(\mathrm{d}k, \mathrm{d}e) \mathbf{1}_{[0, \bar{\lambda}^e]}(\theta) \mathrm{d}\theta$, and $Q^W$ a random Poisson measure on $\mathbb{R}^+ \times \mathscr{J}_W \times \mathbb{N} \times \mathbb{R}^+$, of intensity $\mathrm{d}t \delta_{\mathscr{J}_{\mathscr{E}}}(\mathrm{d}k, \mathrm{d}e) \delta_{\mathbb{N}}(\mathrm{d}j) \mathbf{1}_{[0, \bar{W}^e]}(\theta) \mathrm{d}\theta$. Finally, let $Q^{\mathscr{P}}$ be a random Poisson measure on $\mathbb{R}^+ \times \mathscr{P} \times \mathbb{R}^+$, of intensity $\mathrm{d}t \delta_P(\mathrm{d}e) \mathbf{1}_{[0, \bar{\mu}^e]}(\theta) \mathrm{d}\theta$.*

*There exists a unique measure-valued process $Z$, strong solution on the following SDE driven by Poisson measure:*

$$\begin{aligned}
Z_t = Z_0 &+ \int_0^t \int_{\mathscr{J}_{\mathscr{E}} \times \mathbb{R}^+} \phi^e(s, I_k) \mathbf{1}_{\{k \leq N_{s^-}\}} \mathbf{1}_{\{\theta \leq \lambda_s^e(I_k, Z_{s^-})\}} Q^{\mathscr{E}}(\mathrm{d}s, \mathrm{d}k, \mathrm{d}e, \mathrm{d}\theta) \\
&+ \int_0^t \int_{\mathscr{J}_W \times \mathbb{N} \times \mathbb{R}^+} \phi^e(s, I_k) \mathbf{1}_{\{k \leq N_{s^-}\}} \mathbf{1}_{\{j \leq N_{s^-}\}} \mathbf{1}_{\{\theta \leq W^e(s, I_k, I_j)\}} Q^W(\mathrm{d}s, \mathrm{d}k, \mathrm{d}e, \mathrm{d}j, \mathrm{d}\theta), \qquad (24) \\
&+ \int_0^t \int_{\mathscr{P} \times \mathbb{R}^+} \phi^e(s, I_{s^-}) \mathbf{1}_{\{\theta \leq \mu_s^e\}} Q^{\mathscr{P}}(\mathrm{d}s, \mathrm{d}e, \mathrm{d}\theta),
\end{aligned}$$

*with $I_{s^-}$ an individual taken uniformly in $Z_{s^-}$.*

*Furthermore, the solution of Equation 24 has the same law than the solution of Equation 12.*

The proof of Theorem 8.1 follows the same steps than the proof of Theorem 3.1.

## 8.3 Proof of Theorem 4.1

For ease of notation, we prove the case when $\mathscr{P} = \emptyset$ (there are no events with Poisson intensity).

Let $Z$ be the population process obtained by Algorithm 2 , and $(T_n)_{n \geq 0}$ the sequence of its jump times $(T_0 = 0)$.

**Step 1** Let $T_1$ be the first event time in the population, with its associated marks defining the type $E_1$ of the event and the individual $I_1$ to which this event occurs. By construction, $(T_1, E_1, I_1)$ is characterized by the first jump of:

$$Q^0(\mathrm{d}t, \mathrm{d}k, \mathrm{d}e) = \int_{\mathbb{R}^+} \mathbf{1}_{\{\theta \leq \lambda_t^e(I_k, Z_0)\}} \bar{Q}^0(\mathrm{d}t, \mathrm{d}k, \mathrm{d}e, \mathrm{d}\theta),$$

with $\bar{Q}^0$ the Poisson measure introduced in the first step of the algorithm described in Section 4.2.

Since $T_1$ is the first event time, the population composition stays constant, $Z_t = Z_0$, on $\{t < T_1\}$. In addition, recalling that the first event has the action $\phi^{E_1}(T_1, I_1)$ (see Table 1) on the population $Z$, we

obtain that:

$$\begin{aligned}
Z_{t \wedge T_1} &= Z_0 + \mathbf{1}_{\{t \geq T_1\}} \phi^{E_1}(T_1, I_1) \\
&= Z_0 + \int_0^{t \wedge T_1} \int_{\mathcal{J}_0} \phi^e(s, I_k) Q^0(\mathrm{d}s, \mathrm{d}k, \mathrm{d}e) \\
&= Z_0 + \int_0^{t \wedge T_1} \int_{\mathcal{J}_0} \int_{\mathbb{R}^+} \phi^e(s, I_k) \mathbf{1}_{\{\theta \leq \lambda_s^e(I_k, Z_0)\}} \bar{Q}^0(\mathrm{d}s, \mathrm{d}k, \mathrm{d}e, \mathrm{d}\theta).
\end{aligned}$$

Since $Z_{s^-} = Z_0$ on $\{s \leq T_1\}$, the last equation can be rewritten as

$$Z_{t \wedge T_1} = Z_0 + \int_0^{t \wedge T_1} \int_{\mathcal{J}_0} \int_{\mathbb{R}^+} \phi^e(s, I_k) \mathbf{1}_{\{\theta \leq \lambda_s^e(I_k, Z_{s^-})\}} \bar{Q}^0(\mathrm{d}s, \mathrm{d}k, \mathrm{d}e, \mathrm{d}\theta).$$

**Step 2** The population size at the $n$–th event time $T_n$ is $N_{T_n}$. The $(n+1)$–th event type and the individual to which this event occur are thus chosen in the set

$$\mathcal{J}_n := \{1, \ldots, N_{T_n}\} \times (\mathcal{E} \cup \mathcal{E}_W).$$

Conditionally to $\mathcal{F}_{T_n}$, let us first introduce the marked Poisson measure $\bar{Q}^n$ on $[T_n, \infty) \times \mathcal{J}_n \times \mathbb{R}^+$, of intensity:

$$\begin{aligned}
\bar{\mu}^n(\mathrm{d}t, \mathrm{d}k, \mathrm{d}e, \mathrm{d}\theta) &:= \mathbf{1}_{\{t > T_n\}} \bar{\Lambda}(N_{T_n}) \mathrm{d}t \frac{\bar{\lambda}_n^e}{\bar{\Lambda}(N_{T_n})} \delta_{\mathcal{J}_n}(\mathrm{d}k, \mathrm{d}e) \frac{1}{\bar{\lambda}_n^e} \mathbf{1}_{[0, \bar{\lambda}_n^e]}(\theta) \mathrm{d}\theta, \\
&= \mathbf{1}_{\{t > T_n\}} \mathrm{d}t \, \delta_{\mathcal{J}_n}(\mathrm{d}k, \mathrm{d}e) \mathbf{1}_{[0, \bar{\lambda}_n^e]}(\theta) \mathrm{d}\theta,
\end{aligned} \tag{25}$$

with $\bar{\lambda}_n^e = \bar{\lambda}^e \mathbf{1}_{e \in \mathcal{E}} + \bar{W}^e N_{T_n} \mathbf{1}_{e \in \mathcal{E}_W}$.

By definition, $\bar{Q}^n$ has no jump before $T_n$.

As for the first event, the triplet $(T_{n+1}, E_{n+1}, I_{n+1})$ is determined by the first jump of the measure $Q^n(\mathrm{d}s, \mathrm{d}k, \mathrm{d}e) := \int_{\mathbb{R}^+} \mathbf{1}_{\{\theta \leq \lambda_s^e(I_k, Z_{T_n})\}} \bar{Q}^n(\mathrm{d}s, \mathrm{d}k, \mathrm{d}e, \mathrm{d}\theta)$, obtained by thinning of $\bar{Q}^n$. Finally, since the population composition is constant on $[T_n, T_{n+1}[$, $Z_t = Z_{T_n}$, the population on $[0, T_{n+1}]$ is defined by:

$$\begin{aligned}
Z_{t \wedge T_{n+1}} &= Z_{t \wedge T_n} + \mathbf{1}_{\{t \geq T_{n+1}\}} \phi^{E_{n+1}}(T_{n+1}, I_{n+1}), \\
&= Z_{t \wedge T_n} + \int_{t \wedge T_n}^{t \wedge T_{n+1}} \int_{\mathcal{J}_n \times \mathbb{R}^+} \phi^e(s, I_k) \mathbf{1}_{\{\theta \leq \lambda_s^e(I_k, Z_{s^-})\}} \bar{Q}^n(\mathrm{d}s, \mathrm{d}k, \mathrm{d}e, \mathrm{d}\theta).
\end{aligned} \tag{26}$$

Applying $n$ times Equation 26 yields that:

$$Z_{t \wedge T_{n+1}} = Z_0 + \sum_{l=0}^n \int_{t \wedge T_l}^{t \wedge T_{l+1}} \int_{\mathcal{J}_l \times \mathbb{R}^+} \phi^e(s, I_k) \mathbf{1}_{\{\theta \leq \lambda_s^e(I_k, \tilde{Z}_{s^-})\}} \bar{Q}^l(\mathrm{d}s, \mathrm{d}k, \mathrm{d}e, \mathrm{d}\theta).$$

**Step 3** Finally, let $\tilde{Z}$ be the solution of Equation 12, with $(\tilde{T}_n)_{n \geq 0}$ the sequence of its event times. Then, we can write similarly for all $n \geq 0$:

$$\begin{aligned}
\tilde{Z}_{t \wedge \tilde{T}_{n+1}} &= Z_0 + \sum_{l=0}^n \int_{t \wedge \tilde{T}_l}^{t \wedge \tilde{T}_{l+1}} \int_{\mathcal{J} \times \mathbb{R}^+} \phi^e(s, I_k) \mathbf{1}_{\{\theta \leq \lambda_s^e(I_k, \tilde{Z}_{s^-})\}} \mathbf{1}_{\{k \leq \tilde{N}_{s^-}\}} Q(\mathrm{d}s, \mathrm{d}k, \mathrm{d}e, \mathrm{d}\theta), \\
&= Z_0 + \sum_{l=0}^n \int_{t \wedge \tilde{T}_l}^{t \wedge \tilde{T}_{l+1}} \int_{\mathcal{J} \times \mathbb{R}^+} \phi^e(s, I_k) \mathbf{1}_{\{\theta \leq \lambda_s^e(I_k, \tilde{Z}_{s^-})\}} \mathbf{1}_{\{\theta \leq \tilde{\lambda}_n^e\}} \mathbf{1}_{\{k \leq \tilde{N}_{\tilde{T}_l}\}} Q(\mathrm{d}s, \mathrm{d}k, \mathrm{d}e, \mathrm{d}\theta),
\end{aligned}$$

since $\tilde{Z}_{s^-} = \tilde{Z}_{T_l}$ on $]\tilde{T}_l, \tilde{T}_{l+1}]$, and

$$\lambda_s^e(I_k, \tilde{Z}_{s^-}) \leq \tilde{\lambda}_n^e := \bar{\lambda}^e \mathbf{1}_{e \in \mathcal{E}} + \bar{W}^e \tilde{N}_{\tilde{T}_n} \mathbf{1}_{e \in \mathcal{E}_W}$$

For each $l \geq 0$, let

$$\tilde{Q}^l(\mathrm{d}t, \mathrm{d}k, \mathrm{d}e, \mathrm{d}\theta) = \mathbf{1}_{\{t > \tilde{T}_l\}} \mathbf{1}_{\{1,\ldots,\tilde{N}_{\tilde{T}_l}\}}(k) \mathbf{1}_{[0,\tilde{\lambda}_n^e]}(\theta) Q(\mathrm{d}t, \mathrm{d}k, \mathrm{d}e, \mathrm{d}\theta).$$

By Proposition 8.1, $\tilde{Q}^l$ is, conditionally to $\mathscr{F}_{T_l}$, a Poisson measure of intensity

$$\mathbf{1}_{\{t > \tilde{T}_l\}} \mathrm{d}t \mathbf{1}_{\{1,\ldots,\tilde{N}_{\tilde{T}_l}\}}(k) \delta_{\mathscr{F}}(\mathrm{d}k, \mathrm{d}e) \mathbf{1}_{[0,\tilde{\lambda}_n^e]}(\theta) \mathrm{d}\theta.$$

It follows easily by induction that $\tilde{Q}^l$ has thus the same distribution than $\bar{Q}^l$, the Poisson measure with the conditional intensity $\bar{\mu}^l$ defined in Equation 25. Thus, $Z$ is an exact simulation of Equation 12.

## 8.4 Acknowledgements

The research of Sarah Kaakai is funded by the European Union (ERC, SINGER, 101054787). Views and opinions expressed are however those of the author(s) only and do not necessarily reflect those of the European Union or the European Research Council. Neither the European Union nor the granting authority can be held responsible for them.

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

# Session information

```
sessionInfo()
```

```
R version 4.4.1 (2024-06-14)
Platform: aarch64-apple-darwin20
Running under: macOS Sonoma 14.6.1

Matrix products: default
BLAS:   /Library/Frameworks/R.framework/Versions/4.4-arm64/Resources/lib/libRblas.0.dylib
LAPACK: /Library/Frameworks/R.framework/Versions/4.4-arm64/Resources/lib/libRlapack.dylib;  LAPACK

```

```
locale:
[1] en_US.UTF-8/en_US.UTF-8/en_US.UTF-8/C/en_US.UTF-8/en_US.UTF-8

time zone: Europe/Paris
tzcode source: internal

attached base packages:
[1] stats     graphics  grDevices utils     datasets  methods   base

other attached packages:
[1] reshape2_1.4.4  StMoMo_0.4.1     forecast_8.23.0 gnm_1.1-5
[5] IBMPopSim_1.0.0 ggplot2_3.5.1

loaded via a namespace (and not attached):
 [1] dotCall64_1.1-1        gtable_0.3.5           spam_2.10-0
 [4] xfun_0.47              lattice_0.22-6         tzdb_0.4.0
 [7] quadprog_1.5-8         vctrs_0.6.5            tools_4.4.1
[10] generics_0.1.3         curl_5.2.3             parallel_4.4.1
[13] tibble_3.2.1           fansi_1.0.6            xts_0.14.0
[16] pkgconfig_2.0.3        Matrix_1.7-0           checkmate_2.3.2
[19] RColorBrewer_1.1-3     lifecycle_1.0.4        rootSolve_1.8.2.4
[22] farver_2.1.2           stringr_1.5.1          compiler_4.4.1
[25] fields_16.2            tinytex_0.53           munsell_0.5.1
[28] htmltools_0.5.8.1      maps_3.4.2             yaml_2.3.10
[31] pillar_1.9.0           MASS_7.3-61            nlme_3.1-166
[34] fracdiff_1.5-3         tidyselect_1.2.1       fanplot_4.0.0
[37] digest_0.6.37          stringi_1.8.4          dplyr_1.1.4
[40] labeling_0.4.3         qvcalc_1.0.3           tseries_0.10-58
[43] RcppArmadillo_14.0.2-1 fastmap_1.2.0          grid_4.4.1
[46] colorspace_2.1-1       cli_3.6.3              magrittr_2.0.3
[49] relimp_1.0-5           utf8_1.2.4             readr_2.1.5
[52] withr_3.0.1            scales_1.3.0           backports_1.5.0
[55] TTR_0.24.4             rmarkdown_2.28         quantmod_0.4.26
[58] nnet_7.3-19            timeDate_4041.110      zoo_1.8-12
[61] hms_1.1.3              urca_1.3-4             evaluate_1.0.0
[64] knitr_1.48             lmtest_0.9-40          viridisLite_0.4.2
[67] rlang_1.1.4            Rcpp_1.0.13            glue_1.7.0
[70] jsonlite_1.8.9         plyr_1.8.9             R6_2.5.1
```

