# OpenReview forum: "Efficient simulation of individual-based population models"
_Computo — Accepted by Computo_

### Review · Reviewer_e6pC · 2024-06-30

**Summary Of Contributions:**

This manuscript is not a scientific study, but the description of a software package that can be used to carry out stochastic Individual-Based Models (IBMs) in various fields. It explains the mathematical framework and also examples of how to implement the models in r-code. It shows how to reproduce the results of other software.

As this is not my field of expertise, I have not reviewed the mathematical claims in depth, nor have a used the code on my own computer. I assume those are correct.

**Audience:**

Yes

**Broader Impact Concerns:**

I have no broader impact concerns.

**Claims And Evidence:**

Yes

**Requested Changes:**

I liked that there first was a short R-example in section 2.1.1-2.1.2, and then the mathematics more described more in detail. However, maybe section 2.2-2.5 should be a new section 3 with a title such as ‘Mathematical framework of IBMPopSim.’

For length concerns, maybe some of the content could be in the package vignette only, for example sections 4-6 that contain R-code. But this of course depends on the Computo journal preferences.

Remove extraneous spaces before or after words throughout the manuscript, for example in lines 665-668.

Specific comments:
L. 56. No new paragraph.
63. New paragraph.
71. ‘Over long time scales’
76. Remove ‘often referred to’.
92. This sentence is confusing. Are users required to use C++ code? Maybe: ‘IBMPopSim offers user-friendly R as it generates the C++ code’?
95. “Subsequently, Rcpp is invoked to compile the model so that the model is integrated it into the R session and callable with varying parameters, enabling the generation of diverse population evolution scenarios”.
105. The relationship of this paragraph to the others is unclear. Add an introductory paragraph, like ‘MicSim is a similar package to generate IDM models, but uses microsimulation which is slow….’.
105. Avoid repetition: ‘The R package MicSim (Zinn 2014) can be used for continuous time microsimulation, in which individual life-courses…‘
132. ‘population evolution’ (no plural)
132. Microscopic is only used once in the article, maybe remove, or explain.
142. Maybe interactions should be added to this list?
143. ‘This is the case for entry events when a new individual joins the population, which is when the model specifies how the age and characteristics of this new individual are chosen.’
168. ‘Function’ in singular.
168. To clarify model structure, maybe add: ’In this example, the mk_event_individual function is used by the mk_model function.’ Or add elsewhere ‘All events are run in the mk_model function’.
254. ‘…each event e occurs in the population has to be specified’.
289. ‘Classic case’.
293. ‘Its size x.’
294. ‘then we have’
308. ‘computationally costly’.
308. Remove repeated algorithm.
312. ‘Requires us to…’
314 Should Equation 3 be in parenthesis?
331-332. This sentence is unclear. Should it be ‘its dynamics as a Stochastic…’
233. ‘Pathwise representation of…’
334. Move the parenthesis to the publication year.
334. remove ‘in many examples’.
339. ‘representations of ‘
347. I think it should be ‘non-explosion’ with a line inbetween.
362. What is ‘en’?
574. ‘to automatically generate’.
594. ‘will contain’
667. Unclear, rewrite ‘where we observe the gain of a factor of 40 between the two algorithms, on a set of standard parameters.’ Maybe: ‘algorithm X was 40 times faster than algorithm Y.’
669. Rewrite ‘This allows in particular to explore parameter sets that give larger population sizes, without reaching computation times that explode.’. Maybe ‘… which allow the exploration of larger population sizes with reaching long computation times’.

**Strengths And Weaknesses:**

The manuscript structure is ok, but I provide some suggestions for improvement below. The first paragraphs could be restructured somewhat for clarity. The English language is good, I have some comments below.

---

> ### Comment · Reviewer_e6pC · 2024-06-30
> **I think this manuscript, with some reorganizations and corrections of the text, would be a adequate contribution to Computo.**
>
> I think this manuscript, with some reorganizations and corrections of the text, would be a adequate contribution to Computo.

---

> ### Author Response · Authors · 2024-10-11
> **Response to the reviewer e6pC**
>
> Dear reviewer and editors,
>
>
> We thank the referee for the report. We have addressed all the points raised in the report and made changes in the paper accordingly.
> Please find below a more detailed answer to some of the comments:
>
> - I liked that there first was a short R-example in section 2.1.1-2.1.2, and then the mathematics more described more in detail. However, maybe section 2.2-2.5 should be a new section 3 with a title such as ‘Mathematical framework of IBMPopSim.’ : We have reorganized the paper as suggested and changed the introduction.
>
> - For length concerns, maybe some of the content could be in the package vignette only, for example sections 4-6 that contain R-code. But this of course depends on the Computo journal preferences : we leave this point to the editor.
>
> - Remove extraneous spaces before or after words throughout the manuscript, for example in lines 665-668. :  The extra spaces after the algorithm references are due to the quarto style.
>
> Specific comments:
>
> -  92. This sentence is confusing. Are users required to use C++ code? Maybe: ‘IBMPopSim offers user-friendly R as it generates the C++ code’?  :  For clarity, we modified the beginning of the sentence.
>
> - 105. The relationship of this paragraph to the others is unclear. Add an introductory paragraph, like ‘MicSim is a similar package to generate IDM models, but uses microsimulation which is slow….’.  105. Avoid repetition: ‘The R package MicSim (Zinn 2014) can be used for continuous time microsimulation, in which individual life-courses…’  : We have modified the paragraph and hope to have improved clarity.
>
> - 132. Microscopic is only used once in the article, maybe remove, or explain.  : microscopic has been changed to individual.
>
> - 142. Maybe interactions should be added to this list? Interactions are not considered as a type of event. We say that an event (whatever its type) has interaction if the intensity at which this event can occur to an individual depends not only on the individual’s characteristics, but also on the population composition :   This point has been clarified at the end of the paragraph, on  line 155 in the new version.
>
> - 168. ‘Function’ in singular.  :  sorry, we did not understand this comment.
>
> - 168. To clarify model structure, maybe add: ’In this example, the mk_event_individual function is used by the mk_model function.’ Or add elsewhere ‘All events are run in the mk_model function’.  : We have rewritten the sentence which is now on line 167.
>
> - 314 Should Equation 3 be in parenthesis? We used the quarto cross referencing style, which does not provide parenthesis for equation references (if we are not mistaken).
>
> - 334. remove ‘in many examples’ : This representation has actually been proved  only in particular case, for various models. We provide here a general mathematical framework.
>
> -  362. What is ‘en’? The statement of assumption 2.4 has been clarified.
>
> -  669. Rewrite ‘This allows in particular to explore parameter sets that give larger population sizes, without reaching computation times that explode.’. Maybe ‘… which allow the exploration of larger population sizes with reaching long computation times’. : For ease of reading,  we have rewritten the sentence.

---

> ### Comment · Reviewer_e6pC · 2024-10-29
> **Comments on revision**
>
> It seems like the authors have responded to all my comments. I can recommend this submission for publication.
>
> Comments.
> Original line 334. remove ‘in many examples’ : This representation has actually been proved only in particular case, for various models. We provide here a general mathematical framework.
>
> : Then write “actually been proved only in particular case, for various models. We provide here a general mathematical framework.” instead. “in many examples” sounds vague.

---

> > ### Author Response · Authors · 2024-11-05
> > **Changes to comment done**
> >
> > Dear reviewer and editors,
> > We thank you for your comments and your recommendation.
> > We have made the suggested modifications.
> > Kind regards

---

### Review · Reviewer_AxD8 · 2024-09-16

**Summary Of Contributions:**

This manuscript presents and details the features of the R package IBMPopSim.

This package aims at simulating efficiently (therefore very rapidly) populations modeled by birth-and-death processes with interactions, immigration, and structure, or individual characteristics.

More precisely, individuals are characterized by their age, and some characteristics (for instance position, genome, risk level,...). Each individual can reproduce, die, or change characteristics, at rates that can depend both on its characterics and age, and on the characteristics and age of other individuals of the population. Some individuals can also immigrate in the population, according to a Poisson process. All these events are modeled using a measure-valued population process that satisfies a stochastic differential equation driven by a (possibly complicated) Poisson measure (Theorem 3.1).


The efficiency of the algorithms implemented in IBMPopSim relies on the thinning of Poisson measures : all birth, death, immigration, changes in characteristics rates are assumed to be bounded, which allows to generate a maximum amount of events, some of which being rejected when simulating the considered population. In fact two algorithms are implemented in IBMPopSim, respectively based on deterministic and random thinning of Poisson measures. The random thinning allows a division by $19$ of the duration of simulations, which can be essential when using bayesian estimation methods to estimate parameters, for instance.

**Audience:**

Yes

**Broader Impact Concerns:**

No concern

**Claims And Evidence:**

Yes

**Requested Changes:**

Questions :

1- I could not see why the standard algorithm (Algorithm 2) is useful (and why at least you put it as the "by default" configuration), if the random one (Algorithm 3) is much faster. Maybe you said it somewhere, I cannot remember at this stage.

2- Does/can the simulation stop when it reaches an absorbing state ? Typically in a lot of situations it can be very useful/efficient to stop the simulation when the population is extinct.

3- Can the immigration be other than Poissonian ? Or could you give some advice on how to offer this possibility of generalization ? This also could be very useful, as immigration can in fact result from very diverse ecological/economical processes.

4- In the examples that you give the populations are very large (which is obviously relevant to prove the usefulness of your package). Could you assume in one of these examples that the dynamics of the population, once properly rescaled, is close to a dynamical system, or to a stochastic diffusion, as is often the case for birth-and-death processes ? If so these limiting processes are probably more rapid to simulate than the exact birth-and-death process. I think you should comment a little bit on that, or maybe show that you package allows to consider more general situations, maybe involving multi-scale situations, or explain that the cases that you consider do not correspond to these particular scalings.

Typos :

1- First line of the abstract : I would write "at simulating" (but I am not an english native speaker)

2- l. 202 : remove "Section"

3- a space is missing on l. 249, 346, 373, 402, 675

4- l. 265 : remove "to"

5- l. 278 : "\textbf{an} event of type $e$ occur\textbf{s}" instead of "the event of type $e$ occur"

6- l. 296 : depends

7- l. 299, 300 : I would write "it/its" for the individual, and not "he/his"

8- there is an excessive space on l. 311, 471, 568, 570, 668, 669, 949, 1046. I am not confident I gave all of them, it seems that you generally have a problem after citations.

9- l. 317 : put "equation 3" between parentheses

10- l. 323: remove "i.e. $\lambda^e$ verifies" and put "Equation 5" between parentheses

11- l. 343: you could cite part of this abundant literature

12- l. 355: you could write "(i.e. when $W^b(t,I,J)=C_b$)" to make a distinction with the equation just before

13- l. 397 \textbf{relies} on \textbf{a} pathwise

14- l. 417 : implemented algorithm

15- l. 459 : remove "the"

16- first line of algorithm 1 : shouldn't $\bar{\lambda}$ be $\bar{\lambda_j}$ ?

17- l. 502 : you could write $j=(k,e)$ at this line instead of line 505.

18- l. 569 : put "an event with interaction" between parentheses"

19- l. 624 : remove Section

20- l. 697 : specifying

21- l. 706: its

22- l. 776 : given

23- l. 840 : functions

24- l. 869 : inherits

25- l. 964 : If an event

26- l. 980 : remove the coma

27- l. 986, 959 : straightforward

28- l. 1043 : The proof of

29- l. 1061: jump

30- l. 1072 : is an exact simulation

**Strengths And Weaknesses:**

This R Package should be extremely useful for researchers working in probability applied to ecology, genetic evolution, and finance, as is mentioned in several examples described in the article. The package seems also quite easy to use. Overall I think this package is very useful and that the associated article is very well presented and rigorous.

---

> ### Author Response · Authors · 2024-10-11
> **Response to the reviewer AxD8**
>
> Dear reviewer and editors,
>
> We have taken into account the reviewer’s remarks and modified the paper accordingly.
> In addition, some minor modifications have also been made to the package, as a result of the reflections that followed the review. We have uploaded a new version of the package on github, available now, and a version on CRAN, currently being registered.
>
> Please find below a more detailed answer to some of the questions:
>
> 1- We chose to present the standard algorithm (Algorithm 2) first, without randomization, as it is the most commonly used approach. Indeed, when interactions are quadratic (Equation 6), as in the context of this paper, the randomized approach (Algorithm 3) is often much faster.
> In the package, it is actually the ‘random’ algorithm which has been set as the default algorithm (see for instance the paragraph  "Event creation with interaction intensity'' in Section 5.2.1).
>
> 2- Indeed, the simulation terminates when the population goes extinct and no immigration events occur. However, if the population is empty and immigration follows a Poisson process, the simulation will continue. An additional check has been implemented to stop the simulation earlier when the global upper bound of the system’s intensity rates is zero.
>
> 3- In the package, immigration/entry events (or actually any event) can occur with a constant intensity, corresponding to the jump times of a Poisson process. Alternatively, this intensity may vary with time, corresponding to the jump times of an inhomogeneous Poisson process. Such events can be generated using the "mk_event_inhomogeneous_poisson" function, which facilitates the incorporation of immigration events that exhibits seasonal intensity patterns for instance.
>
> More generally, the intensity can be influenced by various exogenous random processes representing the environment. In this scenario, the event times correspond to the jump times of a Cox process. The conditional intensity can be simulated first and then defined in the "intensity" variable of the "mk_event_inhomogeneous_poisson" function.
>
> 4- Thanks for your interesting comment. In in the examples provided in the package, and more generally under the assumptions of the paper, a large population approximation can indeed be derived for the rescaled microscopic process. At the limit, the rescaled IBM can be approximated by a non linear transport PDE, structured by age and trait. A central limit theorem can also be obtained under appropriate assumptions (Tran (2008)).
>
> It can be noted that in the presence of interactions as in Section 7 for instance, the IBM goes almost surely to extinction in finite time, which is not the case for the limit PDE. In this case, simulating the microscopic process can be quite useful for approximating the distribution of the extinction time. Other applications of the package can indeed include the simulation of multiscale population evolution or strongly heterogeneous populations, or small populations with strong (and thus computationally intensitive) interactions.
>
> From a statistical viewpoint, simulating a large number of population path samples can be interesting to generate synthetic data and/or to compute empirical confidence intervals.
>
> Finally, since the limit equation is structured by age and trait, it can be quite long in certain cases to approximate the limit PDE, especially when the space of traits is large. In such case, approximating the limit PDE using the rescaled microscopic process might be efficient.
>
> Following your suggestion we have clarified this point in the paper on page 13 at the beginning of Section 4.
>
>
> Typos
>
> 1.  We change the first line of abstract: The R package IBMPopSim facilitates the simulation of the random evolution of heterogeneous populations through stochastic Individual-Based Models.
> 2.  done
> 3.  done
> 4.  the sentence has been refined
> 5.  done
> 6.  done
> 7.  done
> 8.  The spaces present in the PDF when referencing algorithms are automatically added during the compilation phase with Quarto. These spaces are not present in the HTML version, but only in the pdf output. We believe this issue may require a modification to the Quarto stylesheet used by the Computo journal. Thus, we did not apply a manual correction, and we leave it to the editor to decide on the best solution.
> 9.  done
> 10. done
> 11. We have added some references.
> 12. done
> 13. done
> 14. done
> 15. done
> 16. yes, this has been corrected.
> 17. yes, this has been corrected.
> 18. done
> 19. done
> 20. done
> 21. done
> 22. done
> 23. l. 846 done
> 24. done
> 25. done
> 26. done
> 27. done
> 28. done
> 29. done
> 30. done

---

### Author Response · Authors · 2024-07-08
**Questions sur le processus de révision**

Bonjour,

Nous avons bien pris note du premier rapport soumis le 30 juin. Nous nous interrogeons sur le processus de révision. Doit-on attendre un autre rapport et/ou avis des éditeurs, ou doit-on faire les corrections demandées dans un premier temps?

Bien cordialement,

D. Giorgi, S. Kaakai et V. Lemaire.

---

### Comment · Action_Editor_QZjq · 2024-07-12
**Réponse sur le processus de révision**

Bonjour,

Désolée pour le délai de réponse !

Vous pouvez commencer à répondre au première relecteur sans attendre les autres rapports. Le processus de revision avec open review se fait en plusieurs étapes, avec une étape d'échange avec les rapporteurs avant la décision éditoriale (comme cela peut être le cas pour une conférence par exemple )

Merci,
Nelle

---

> ### Author Response · Authors · 2024-07-12
> **Réponse au premier relecteur**
>
> Bonjour,
> Nous avons sousmis une revision du papier, en suivant les indications du premier relecteur, et en mettant dans le commentaire "Changes since last submission" notre réponse au premier relecteur.
> Cordialement
> D. Giorgi, S. Kaakai et V. Lemaire.

---

### Author Response · Authors · 2024-10-11
**Cleaning step**

We've moved the reply to the first reviewer from the “Changes from last submission” section to an “Official comment” in the dedicated section.

---

### Comment · Action_Editor_QZjq · 2024-12-03
**Proofreading before publication**

Dear authors,

Your paper is ready for publication at https://computo.sfds.asso.fr/published-202412-giorgi-efficient/ (html version) and https://computo.sfds.asso.fr/published-202412-giorgi-efficient/published-202412-giorgi-efficient.pdf (pdf version)

Should you notice any typo (in particular, in affiliations etc), can you send a PR to https://github.com/computorg/published-2024202412-giorgi-efficient before December 15th, 2024?

Best,

---

> ### Author Response · Authors · 2024-12-10
> **Final revision**
>
> Dear Editors,
>
> We have carefully reviewed the paper and made some minor adjustments, that we pushed on the github repo :
>
> - The affiliation for S. Kaakai has been updated.
> - To ensure compatibility with the HTML version, Figures 1 and 6 have been replaced with PNG files instead of TikZ code.
>
> However, some issues with the HTML rendering persist:
>
> **Major Issues:**
>
> - Algorithm 3 is missing in the HTML version.
> - The formatting of Algorithm 4 is incorrect.
>
> **Minor Remarks:**
>
> - Bold text does not display correctly in the HTML output.
> - The zoom-on-hover feature for citations and references is inconsistent and does not always work.
>
> Do you think that the major issues can be solved before publication ?
> Kind regards
> D. Giorgi, S. Kaakai et V. Lemaire

---

> > ### Comment · Action_Editor_QZjq · 2024-12-16
> > **RE: final revision**
> >
> > Dear authors,
> >
> > We are looking into the issues. They will be fixed before publication (although probably not before the christmas holidays).
> >
> > Many thanks,
> > Nelle

---

> > > ### Comment · Action_Editor_QZjq · 2025-01-27
> > >
> > > Dear authors,
> > >
> > > The problems with the algorithm are now solved. There are still the issues with the zoom-on-hover feature, which is common to several papers. I have passed on  this issue onto the production team.
> > >
> > > Now that the major issues have been fixed, I will go ahead with the publication of the article.
> > >
> > > Many thanks,
> > > Nelle

---

### Note · Reviewer_AxD8 · 2024-11-12

**Comment:**

The authors seem to have responded to all my comments. I can recommend this submission for publication. I think the manuscript and the associated package are very interesting and useful.

**Audience:**

Yes

**Claims And Evidence:**

Yes

**Decision Recommendation:**

Accept

---

### Note · Reviewer_e6pC · 2024-11-20

**Audience:**

Yes

**Claims And Evidence:**

Yes

**Decision Recommendation:**

Leaning Accept

---

### Decision · Action_Editor_QZjq · 2024-12-02

**Recommendation:** Accept as is

**Comment:**

The authors have addressed all the points raised by the reviewers.

**Audience:**

This paper is relevant for computo's readership.

**Claims And Evidence:**

Dear authors,

I am pleased to inform you that your paper is accepted.

---

> ### Decision · Editors_In_Chief · 2024-12-02
>
> I approve the AE's decision.